# SOLIDGEO: Measuring Multimodal Spatial Math Reasoning in Solid Geometry

**Peijie Wang**[*,1,2], **Chao Yang**[*,3], **Zhong-Zhi Li**[1,2], **Fei Yin**[1,2], **Dekang Ran**[1,2]
**Mi Tian**[4], **Zhilong Ji**[4], **Jinfeng Bai**[4], **Cheng-Lin Liu**[†,1,2]

[1]MAIS, Institute of Automation of Chinese Academy of Sciences
[2]School of Artificial Intelligence, University of Chinese Academy of Sciences
[3]University of Electronic Science and Technology of China    [4]TAL

{wangpeijie2023, lizhongzhi2022, randekang2025}@ia.ac.cn
{jizhilong, baijinfeng, tianmi}@tal.com  {fyin, liucl}@nlpr.ia.ac.cn

## Abstract

Geometry is a fundamental branch of mathematics and plays a crucial role in evaluating the reasoning capabilities of multimodal large language models (MLLMs). However, existing multimodal mathematics benchmarks mainly focus on plane geometry and largely ignore solid geometry, which requires spatial reasoning and is more challenging than plane geometry. To address this critical gap, we introduce **SOLIDGEO**, the first large-scale benchmark specifically designed to evaluate the performance of MLLMs on mathematical reasoning tasks in solid geometry. SOLIDGEO consists of 3,113 real-world K–12 and competition-level problems, each paired with visual context and annotated with difficulty levels and fine-grained solid geometry categories. Our benchmark covers a wide range of 3D reasoning subjects such as projection, unfolding, spatial measurement, and spatial vector, offering a rigorous testbed for assessing solid geometry. Through extensive experiments, we observe that MLLMs encounter substantial challenges in solid geometry math tasks, with a considerable performance gap relative to human capabilities on SOLIDGEO. Moreover, we analyze the performance, inference efficiency and error patterns of various models, offering insights into the solid geometric mathematical reasoning capabilities of MLLMs. We hope SOLIDGEO serves as a catalyst for advancing MLLMs toward deeper geometric reasoning and spatial intelligence. The dataset is released at SOLIDGEO.

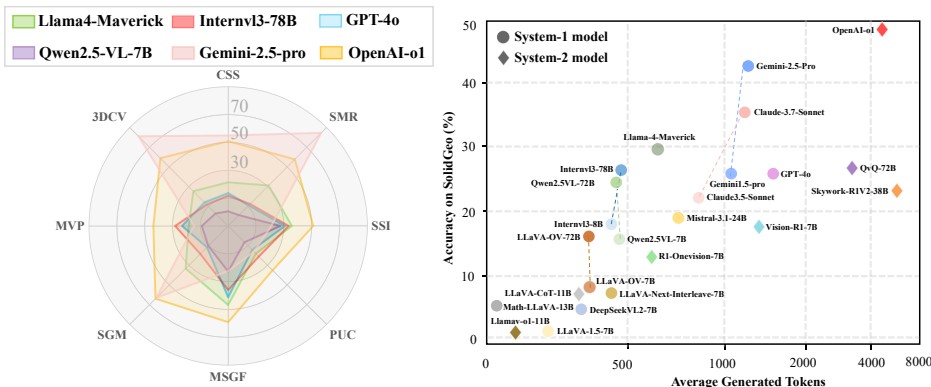

Figure 1: Performance of six MLLMs on SOLIDGEO benchmark across 8 solid geometry subjects (left), and trade-off between accuracy and average generated token length across 25 MLLMs (right).

---

[*] Equal contribution   [†] Corresponding author

39th Conference on Neural Information Processing Systems (NeurIPS 2025) Track on Datasets and Benchmarks.

# 1 Introduction

*"There is no royal road to geometry." — Euclid*

Geometric problems hold a vital position in mathematics and are widely regarded as the foundation and core of mathematics [1, 2]. The challenge of geometric problem solving lies in the integration of complex visual information and symbolic reasoning. Based on the structural properties of geometric figures, geometry can be categorized into plane geometry and solid geometry [3]. Compared to plane geometry, solid geometry involves understanding three-dimensional structures and spatial relationships, making it inherently more complex. Requiring advanced spatial reasoning and the integration of visual and textual modalities, solid geometry represents a particularly challenging class of problems for artificial intelligence systems. [4, 5, 6, 7].

In recent years, advances in Large Language Models (LLMs) [8, 9, 10, 11, 12, 13] and Multimodal Large Language Models (MLLMs) [14, 15, 16, 17, 12, 18] have led to impressive performance across a wide range of language and visual-language tasks, such as natural language understanding, code generation, image captioning and visual question answering [19, 20, 21, 22, 23, 24, 25, 26, 27]. With the growing capabilities of MLLMs, their mathematical reasoning abilities have emerged as a critical focus of recent research. Notably, recent models such as GPT-4o [28], Gemini [29], Qwen-VL [30] and InternVL [16] have surpassed the average human performance on MathVista [31], one of the most widely used benchmarks for multimodal mathematical reasoning.

To better understand the limitations of current multimodal mathematical reasoning benchmarks, we conducted a detailed analysis of existing datasets with a focus on geometry-related content. Our investigation reveals two key issues:

Firstly, although geometry problems are among the most prevalent types of mathematical tasks—as evidenced by the abundance of benchmarks such as Geometry3K [32], GeoQA [33], PGPS9K [34], UniGeo [35] and GeomRel [36]—most existing geometry benchmarks overwhelmingly concentrate on *plane geometry*. In contrast, *solid geometry*, which entails reasoning about three-dimensional structures and their spatial properties such as projection and spatial measurement, remains severely underrepresented. This imbalance persists despite the fundamental role that solid geometry plays in both human curricula and machine reasoning. Crucially, solving solid geometry problems demands advanced spatial reasoning—an essential aspect of spatial intelligence, which has been identified as a key competency on the road toward artificial general intelligence (AGI) [37].

Secondly, although the visual data in existing benchmarks span a wide range of sources and modalities, the scope and depth of solid geometry problems remain notably limited. For instance, MathVista contains only 62 solid geometry questions, all involving simple object counting tasks [31]. MathVision includes 244 such problems [6], and MathVerse provides 119 [7], but we found that the majority focus on recognizing object structures, shapes, or performing basic volume calculations. Critically, more advanced problem types such as projection analysis, spatial transformations, and complex spatial relationship reasoning are rarely represented. This imbalance underscores a significant gap in the current landscape of multimodal mathematical benchmarks: the absence of rich, diverse, and cognitively demanding tasks targeting solid geometry reasoning.

Despite recent progress in multimodal mathematical reasoning, the current benchmarks for solid geometry remain narrow in both topical coverage and problem diversity. As a result, the spatial reasoning capabilities of MLLMs—particularly in 3D geometric contexts—have not been adequately assessed. Solid geometry inherently integrates symbolic mathematical reasoning with spatial intelligence and is regarded as essential for achieving AGI [2, 37]. Yet, this critical domain remains largely overlooked in existing evaluations. To bridge this gap, we introduce the SOLIDGEO dataset. Unlike prior datasets that only sparsely include solid geometry content and focus on basic recognition or counting tasks, SOLIDGEO offers a comprehensive collection of diverse and challenging problems that span a wide range of solid geometry concepts. Our goal is to establish a rigorous benchmark that better reflects the demands of real-world solid geometry reasoning and pushes the frontier of spatial intelligence in multimodal AI systems.

SOLIDGEO comprises solid geometry problems sourced from six existing multimodal math datasets, along with a collection of high-quality questions that we newly curated and annotated from real-world K–12 and high school competition educational scenarios. In total, SOLIDGEO contains **3,113** solid geometry problems, each accompanied by at least one image. Among these, 1,737 samples are derived from existing datasets, while 1,376 are newly collected and verified by our team. The dataset

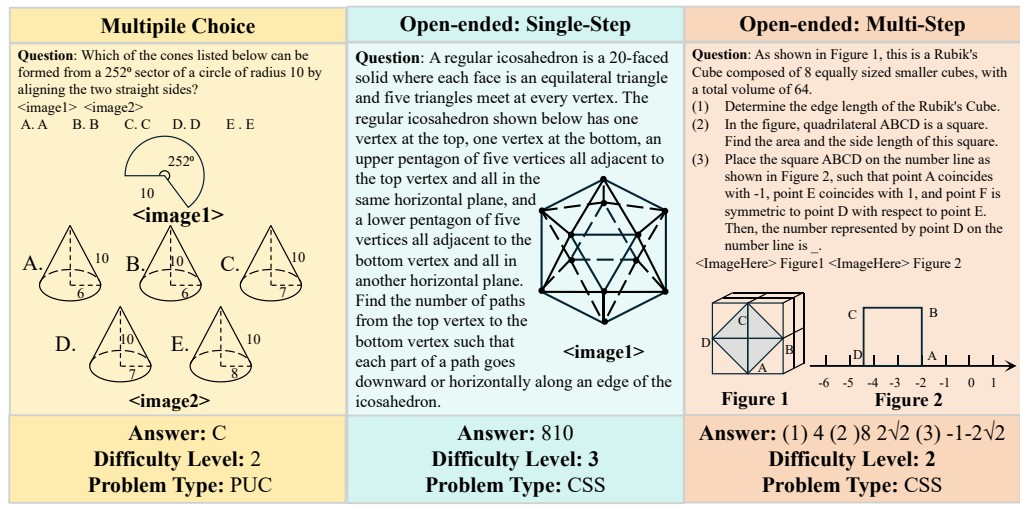

Figure 2: Sampled SOLIDGEO question examples from each question type. Each sample contains a visual context, difficulty levels and fine-grained solid geometry categories.

covers three problem formats: multiple-choice, open-ended single-step, and open-ended multi-step questions. As shown in Figure 2, every question in SOLIDGEO contains at least one image input, annotated with difficulty levels and fine-grained solid geometry categories. Specifically, we introduce three major improvements in SOLIDGEO:

**1. Fine-grained Category.** We present the first fine-grained categorization of solid geometry problems, dividing all questions into eight reasoning-based subcategories. This classification captures core aspects of spatial intelligence and enables a more structured evaluation of model capabilities.

**2. Real-world Problem.** All problems and images in SOLIDGEO are sourced from authentic scenarios, resulting in naturally phrased and diverse questions. Notably, SOLIDGEO has a much higher average question length (77.2) compared to MathVista (15.6) and MathVision (42.3), offering richer contextual information and posing greater challenges.

**3. Difficulty level.** Each problem is labeled with a difficulty level from 1 to 3, verified by domain experts. This enables fine-grained model analysis and helps identify reasoning bottlenecks, particularly for o1-like models whose efficiency may be sensitive to problem complexity.

We conduct extensive experiments on SOLIDGEO to comprehensively assess model performance in solid geometry reasoning tasks. Among the evaluated models, OpenAI-o1 achieves the highest accuracy with a score of 49.5%. Notably, the open-source model Llama 4 [38] performs competitively, scoring 29.6% and outperforming GPT-4o, while ranking just below Claude-3.7-Sonnet. However, all models still fall significantly short of human-level performance. Our analysis provides insights into the strengths and limitations of current open-source MLLMs in handling complex solid geometry spatial reasoning. In summary, our main contributions are as follows:

- We carefully reviewed existing datasets and found that solid geometry problems are insufficiently covered. Despite being crucial for spatial reasoning, a core capability for achieving AGI, solid geometry has been underestimated in previous benchmarks.

- We present **SOLIDGEO**, the first benchmark dedicated to solid geometry mathematical reasoning. It comprises 3,113 problems with visual context, drawn from real-world K–12 and competition sources. Problems are categorized into 3 difficulty levels and 8 fine-grained categories.

- Leveraging fine-grained annotations, we conduct a detailed evaluation of 27 MLLMs, providing insights into their geometric spatial reasoning capabilities and offering analysis of inference efficiency for o1-like models, while identifying key limitations for future research.

## 2 Related Works

**Benchmark for Mathematical Reasoning.** Various benchmarks have been proposed to evaluate the mathematical reasoning capabilities of MLLMs. Early multimodal math datasets such as GEOS [39], GeoQA [33], Geometry3K [32], and UniGeo [35] cover only a narrow range of topics and primarily focus on plane geometry. More recent benchmarks like MMMU [40] include only a small subset of math-related questions and lack any coverage of solid geometry. MathVista [31] includes just 62 solid geometry problems, all of which are basic object counting tasks that do not require complex spatial reasoning. Although MATH-Vision [6], WE-MATH [41], MV-MATH [42], and GeoSense [43] present more diverse and rigorous math problems, they contain only a limited number of solid geometry samples and lack fine-grained categorization. In contrast, SOLIDGEO offers a comprehensive and diverse set of solid geometry problems. It encompasses both visual recognition and complex spatial reasoning, and features explicit difficulty annotations and 8 fine-grained categories—enabling more systematic evaluation of model performance on solid geometry tasks.

**Multimodal Models.** With the advancement of LLMs and vision-language alignment techniques, early multimodal models such as MiniGPT-4 [44], LLaMA-Adapter [45], and LLaVA [14] demonstrated promising capabilities in visual understanding. Recent research also explores parameter-efficient fine-tuning (PEFT) approaches such as LoRA [46] and its improved orthogonal reinitialization strategies UORA [47]. More recent models such as closed-source GPT-4o [28], Claude [48], Gemini [29] and open-source Qwen-VL [30], InternVL [16], and LLaVA-Onevision [49] have further pushed the boundaries of general-purpose visual reasoning. The recent release of OpenAI's o1 model has highlighted the effectiveness of long Chain-of-Thought in improving reasoning performance, this has inspired a wave of o1-style models [50, 51, 52, 53, 54, 55, 56], many of which show strong performance on multimodal mathematical tasks. Beyond step-wise reasoning, Chain-of-Reasoning [57] advocates for a unified multi-paradigm approach, integrating symbolic, visual, and CoT-based reasoning for mathematical problem solving.

Several multimodal models have been specifically developed for mathematical reasoning, including G-LLaVA [58], UniMath [59], LANS [60] and GeoUni [61]. However, these approaches primarily focus on plane geometry tasks. Even AlphaGeometry [2], a geometry model that achieves IMO-level, remains ineffective in handling solid geometry. Despite its importance for spatial reasoning, solid geometry remains an underexplored area for current multimodal models.

## 3 The SOLIDGEO Benchmark

### 3.1 Overview

We present **SOLIDGEO**, a carefully curated benchmark designed to evaluate the multimodal spatial and mathematical reasoning capabilities of foundation models in the domain of solid geometry. Unlike previous benchmarks that primarily emphasize plane geometry or provide only limited coverage of solid geometry, SolidGeo targets the unique challenges of understanding and reasoning over three-dimensional structures and their spatial relationships. Solving such problems requires advanced spatial intelligence such as interpreting projections and analyzing 3D configurations, which are essential for assessing a model's ability to integrate visual perception with symbolic reasoning.

SOLIDGEO comprises 3,113 real-world solid geometry problems, collected from K–12 curricula and high school mathematics competitions. Each problem is paired with at least one image and annotated with a difficulty level ranging from 1 (easy) to 3 (hard). The dataset is categorized into eight fine-grained domains: *Composite Solid Structures*, *Spatial Metric Relations*, *Solid Shape Identification*, *Planar Unfolding and Configuration*, *Measurement of Solid Geometric Forms*, *Solid Geometry Modeling*, *Multi-view Projection*, and *3D Coordinate and Vector Reasoning*. These annotations support more granular analysis of model strengths and weaknesses across different task types and reasoning complexities. Compared to existing multimodal math benchmarks, SOLIDGEO offers a richer, more diverse, and cognitively demanding collection of problems. The average question length is substantially higher, reflecting the depth of contextual information and the complexity of reasoning required. Detailed statistics and coverage of SOLIDGEO are presented in Table 1 and Figure 3.

### 3.2 Data Construction

**Data Collection.** To construct SOLIDGEO, we adopt a hybrid data collection strategy that combines solid geometry problems from existing multimodal mathematical reasoning datasets with newly

| Statistic | Number |
|---|---|
| Total questions | 3,113 |
| - Multiple-choice questions | 969 |
| - Open-ended questions | 2144 |
| -Single-step questions | 1936 |
| -Multi-step questions | 208 |
| Difficulties (Easy: Medium: Hard) | 32%:61%:7% |
| Newly collected questions | 1,376 (44.2%) |
| Existing-dataset questions | 1,737 (55.8%) |
| Newly collected images | 3,555 (66.1%) |
| Existing-dataset images | 1,825 (33.9%) |
| **Language** | |
| - English (EN) | 2,192 (70.4%) |
| - Chinese (CN) | 921 (29.6%) |
| Maximum question length | 679 |
| Maximum answer length | 2833 |
| Average question length | 77.2 |
| Average answer length | 312.2 |

Table 1: Key Statistics of SOLIDGEO.

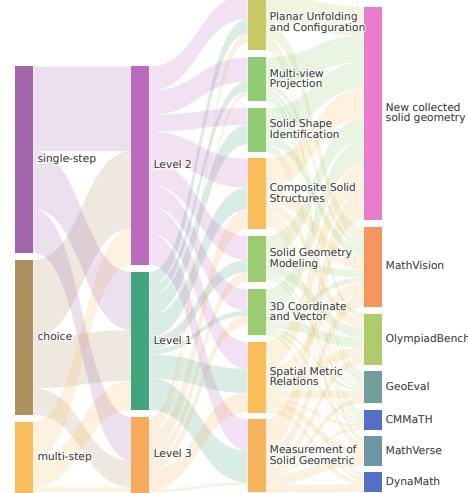

Figure 3: Distribution of SOLIDGEO.

gathered problems from real-world K–12 educational sources. Specifically, we first extracted solid geometry samples from six existing benchmarks based on their provided category labels. To supplement these, we further collected problems from the Zujuan platform, which provides a large-scale repository of K–12 math problems in PDF format. Using the Mathpix API[2], we extracted structured content including questions, answers, analyses and associated diagrams, resulting in an initial pool of 10,932 candidate problems identified via keyword filtering.

**Data Filtering.** To ensure the quality and relevance of the collected solid geometric data, we designed a four-stage filtering pipeline:

- Stage 1: Structural Filtering. We excluded purely textual problems and retained only multimodal samples with at least one associated image. We also removed samples missing essential fields such as answers or diagrams.

- Stage 2: Image Quality Filtering. Using OpenCV, we computed sharpness metrics to eliminate samples containing low-resolution or blurry images, ensuring the retained diagrams were visually clear and suitable for spatial reasoning tasks.

- Stage 3: Semantic Filtering. To refine topic relevance beyond keyword matching, we used the DeepSeek API [13] to classify whether each candidate problem truly belonged to the solid geometry domain, filtering out unrelated samples.

- Stage 4: Cross-Set Deduplication. To prevent redundancy, we applied n-gram [62] based similarity checks between the newly collected samples and those from existing datasets. This ensured that SOLIDGEO contains no near-duplicate or overly similar problems across sources.

Following this pipeline, we retained 1,376 high-quality new problems, which were combined with 1,737 problems from existing datasets, yielding the final SOLIDGEO dataset with 3,113 unique samples. The dataset covers three problem formats: 969 multiple-choice, 1,936 open-ended single-step, and 208 open-ended multi-step questions. Further details on dataset statistics and the filtering process are provided in Appendix A and Appendix D.

**Data Labeling.** The SOLIDGEO dataset incorporates two types of annotations: fine-grained subject categorization and difficulty level assignment. To achieve accurate labeling, we first invited subject-matter experts to define eight representative subjects of solid geometry, covering the full spectrum of spatial reasoning skills required. Each problem was then categorized using a majority voting scheme based on predictions from three advanced MLLMs: GPT-4o, Claude-3.7-Sonnet, and Qwen-VL-Max. Some problems may belong to multiple subjects when applicable. Similarly, difficulty levels ranging from 1 (easy) to 3 (hard) were determined using model-based voting among the same three MLLMs. In cases where consensus could not be reached, final decisions were made by experienced human

---

[2]https://mathpix.com/convert

Table 2: Comparison with existing multimodal math benchmarks. SG: Solid Geometry, PG: Plane Geometry. **Level:** K=K-12, U=University, C=Competition . **Source:** S=Self-sourced, P=Collected from Public Dataset. MC: Multiple Choice, SS: Single-Step, MS:Multi-Step.

| Benchmarks | Language | SG Size | SG Proportion | PG Proportion | Level | Source | SG categroy | MC | SS | MS |
|---|---|---|---|---|---|---|---|---|---|---|
| GeoQA [33] | CN | 0 | 0.0% | 100.0% | K | S | ✗ | ✓ | ✗ | ✗ |
| Geometry3K [32] | EN | 0 | 0.0% | 100.0% | K | S | ✗ | ✓ | ✓ | ✗ |
| UniGeo [35] | EN | 0 | 0.0% | 100.0% | K | S | ✗ | ✗ | ✓ | ✗ |
| PGPS9K [34] | EN | 0 | 0.0% | 100.0% | K | S | ✗ | ✓ | ✓ | ✗ |
| MMMU-MATH [40] | EN | 0 | 0.0% | 29.3% | U | S | ✗ | ✓ | ✓ | ✗ |
| GeoEval [5] | EN | 100 | 2.0% | 94.1% | K | P | ✗ | ✓ | ✗ | ✗ |
| DynaMath [63] | EN | 150 | 3.0% | 15.4% | K | S | ✗ | ✓ | ✓ | ✗ |
| MATH-Vision [6] | EN | 263 | 8.7% | 58.7% | K U | S | ✗ | ✓ | ✓ | ✗ |
| OlympiadBench [64] | EN/CN | 784 | 9.2% | 15.6% | C | S | ✗ | ✗ | ✓ | ✗ |
| MathVerse [7] | EN | 119 | 15.1% | 64.7% | K | S P | ✗ | ✓ | ✗ | ✗ |
| GeoSense [43] | EN/CN | 350 | 20.0% | 80.0% | K | S P | 2 | ✓ | ✓ | ✗ |
| SOLIDGEO (Ours) | EN/CN | 3113 | 100.0% | 0.0% | K C | S P | 8 | ✓ | ✓ | ✓ |

annotators. To ensure annotation quality and consistency, all 3,113 problems finally underwent independent review by three expert annotators, who verified both the subjects and difficulty labels.

## 3.3 Comparison with Existing Benchmarks

Most existing mathematical reasoning benchmarks include only a limited number of solid geometry problems. For example, MathVista [31] contains just 62 solid geometry questions, all following a single templated format: *"Subtract all ... objects. How many objects are left?"*, offering minimal diversity or reasoning depth. MathVerse [7] expands its dataset by restating problems, but its original solid geometry set comprises only 119 unique examples. MathVision [6] provides 244 solid geometry questions, though most of them are concentrated in the domain of structural analysis. Benchmarks such as GeoQA [33], Geometry3K [32], UniGeo [35], and PGPS9K [34] focus almost exclusively on plane geometry, with little or no coverage of 3D reasoning.

In contrast, SOLIDGEO is the first large-scale benchmark dedicated to solid geometry. Each problem is paired with at least one visual input and has been manually verified for correctness. Unlike prior datasets that use broad or ambiguous labels, SOLIDGEO introduces a refined taxonomy with eight fine-grained categories, explicitly capturing the diversity of solid geometry subdomains. Furthermore, SOLIDGEO features a significantly longer average question length (77.2 words), compared to MathVista (15.6) and MathVision (42.3), indicating higher contextual richness and greater reasoning complexity. Together, these attributes make SOLIDGEO a comprehensive and challenging benchmark for evaluating multimodal mathematical reasoning in 3D spatial contexts. A detailed comparison with existing benchmarks is presented in Table 2. See Appendix C for details.

## 4 Experiments

In this section, we conduct a systematic evaluation of existing LLMs and MLLMs on SOLIDGEO. We first introduce the experimental setup in Section 4.1. Then, we detail the quantitative results in Section 4.2 and narrate the error analysis in Section 4.3. The experimental outcomes show that most of the current MLLMs are still far behind the human level in solid geometry.

### 4.1 Experimental Setup

**Evaluation Models.** We evaluate a diverse set of models on SOLIDGEO, spanning both LLMs and MLLMs, including 20 open-source and 7 closed-source models. These models fall into two categories: 19 **System-1 models**, which follow a fast, single-pass reasoning paradigm, and 8 **System-2 models**, which adopt a slow, iterative long CoT reasoning style inspired by o1-type designs [78].

**Implementation Details.** Our evaluation is conducted under three settings: zero-shot direct answering, Chain-of-Thought (CoT), and CoT with 2-shot examples. To establish a human performance baseline, we recruit high school students to independently complete the questions as detailed in Appendix F. For multiple-choice, single-step, and multi-step open-ended formats, we carefully design tailored prompts to ensure models generate responses in the correct structure. To assess the contribution of visual information, we additionally evaluate performance using LLM (text-only) and GPT-4o with text-only inputs. This enables a controlled comparison between purely textual reasoning and full multimodal inputs, thereby quantifying the role of images in solid geometry reasoning. Answer

Table 3: Comparison of model performances across 8 fine-grained solid geometry subjects and average output tokens. CSS: Composite Solid Structures, SMR: Spatial Metric Relations, SSI: Solid Shape Identification, PUC: Planar Unfolding and Configuration, MSGF: Measurement of Solid Geometric Forms, SGM: Solid Geometry Modeling, MVP: Multi-view Projection, 3DCV: 3D Coordinate and Vector Reasoning. The first and second highest accuracy of LMMs are marked in red and blue, respectively.

| Model | Overall | CSS | SMR | SSI | PUC | MSGF | SGM | MVP | 3DCV | Avg.tokens |
|---|---|---|---|---|---|---|---|---|---|---|
| *Text-only, zero-shot direct answering* | | | | | | | | | | |
| Deepseek-V3[13] (LLM) | 9.3 | 10.7 | 8.1 | 8.3 | 12.7 | 6.3 | 7.8 | 10.3 | 12.2 | 787.2 |
| GPT-4o[28] (MLLM) | 9.1 | 10.0 | 10.4 | 10.6 | 6.8 | 12.1 | 8.6 | 7.3 | 9.6 | 692.6 |
| *Open-source MLLMs (Text + Image, zero-shot direct answering)* | | | | | | | | | | |
| *System-1 Models* | | | | | | | | | | |
| LLaVA-v1.5-7B[14] | 1.8 | 1.1 | 1.1 | 6.7 | 2.2 | 0.6 | 0.0 | 4.6 | 0.0 | 246.2 |
| InternLM-XComposer2.5-VL-7B[65] | 4.4 | 2.5 | 1.8 | 6.7 | 8.9 | 0.6 | 0.0 | 9.4 | 1.2 | 151.8 |
| DeepSeek-VL2-7B[66] | 5.1 | 2.8 | 2.6 | 11.1 | 5.1 | 1.4 | 1.8 | 11.7 | 1.8 | 338.2 |
| Math-LLaVA-13B[67] | 5.9 | 4.2 | 4.1 | 7.6 | 11.7 | 2.7 | 4.2 | 12.6 | 6.2 | 7.4 |
| LLaVA-NeXT-Interleave-7B[68] | 7.7 | 2.5 | 2.5 | 21.5 | 13.5 | 2.3 | 7.3 | 16.7 | 0.6 | 486.3 |
| LLaVA-OneVision-Chat-7B[49] | 8.6 | 4.3 | 2.9 | 19.3 | 15.2 | 3.5 | 6.4 | 17.9 | 0.0 | 353.2 |
| Qwen2.5-VL-Instruct-7B[30] | 15.5 | 8.4 | 8.8 | 30.1 | 13.3 | 26.2 | 16.2 | 15.2 | 10.2 | 490.2 |
| LLaVA-OneVision-Chat-72B[49] | 15.9 | 13.2 | 9.5 | 31.9 | 18.1 | 12.9 | 11.8 | 23.7 | 8.4 | 396.3 |
| InternVL3-8B[16] | 17.7 | 11.8 | 10.0 | 24.4 | 17.4 | 28.0 | 19.1 | 19.9 | 7.2 | 488.8 |
| Mistral-small-3.1-24b-instruct[69] | 19.6 | 15.2 | 15.8 | 27.4 | 17.1 | 28.9 | 10.9 | 17.0 | 16.8 | 769.7 |
| Qwen2.5-VL-Instruct-72B[30] | 24.2 | 19.7 | 18.8 | 29.6 | 21.5 | 35.4 | 16.4 | 22.5 | 18.0 | 485.0 |
| InternVL3-78B[16] | 26.2 | 17.4 | 17.9 | 34.8 | 24.9 | 36.8 | 22.7 | 30.5 | 17.4 | 493.2 |
| Llama-4-Maverick-17B-128E[38] | 29.6 | 25.1 | 30.9 | 34.6 | 20.5 | 43.4 | 32.6 | 20.7 | 26.3 | 605.6 |
| *System-2 Models* | | | | | | | | | | |
| LlamaV-o1-11B[70] | 1.5 | 0.6 | 0.7 | 1.5 | 0.5 | 5.0 | 2.7 | 0.1 | 0.0 | 106.1 |
| LLaVA-CoT-11B[71] | 7.3 | 4.2 | 2.5 | 7.4 | 6.5 | 15.1 | 8.2 | 7.4 | 1.8 | 401.7 |
| VLM-R1-3B[72] | 9.6 | 6.3 | 4.4 | 11.1 | 8.7 | 19.6 | 4.5 | 8.3 | 2.4 | 453.0 |
| R1-Onevision-7B[73] | 13.2 | 7.7 | 9.7 | 25.2 | 10.1 | 23.3 | 11.8 | 12.3 | 9.0 | 522.3 |
| Vision-R1-7B[74] | 18.1 | 11.7 | 11.3 | 28.6 | 17.8 | 26.9 | 13.9 | 19.3 | 12.0 | 1498.7 |
| Skywork-R1V2-38B[75] | 23.0 | 18.4 | 29.5 | 13.3 | 11.6 | 31.2 | 30.0 | 12.3 | 26.9 | 5682.9 |
| QvQ-72B-Preview[76] | 26.6 | 17.9 | 28.1 | 37.0 | 22.9 | 34.7 | 20.9 | 20.3 | 22.8 | 3622.2 |
| *Closed-source MLLMs (Text + Image, zero-shot direct answering)* | | | | | | | | | | |
| *System-1 Models* | | | | | | | | | | |
| Claude-3.5-Sonnet[48] | 22.2 | 16.9 | 9.8 | 42.2 | 24.2 | 36.5 | 25.5 | 23.5 | 9.6 | 992.1 |
| GPT-4V[15] | 25.3 | 16.6 | 15.8 | 35.6 | 21.5 | 41.5 | 25.5 | 25.9 | 18.0 | 1433.5 |
| Gemini-1.5-pro[29] | 25.3 | 18.5 | 16.8 | 34.8 | 19.6 | 41.6 | 17.3 | 25.6 | 19.2 | 1003.5 |
| GPT-4o[28] | 25.5 | 18.9 | 16.8 | 32.6 | 19.6 | 41.0 | 17.3 | 26.5 | 19.2 | 1344.9 |
| Claude-3.7-Sonnet[48] | 34.1 | 27.7 | 28.2 | 43.0 | 32.9 | 46.8 | 43.6 | 28.5 | 26.3 | 1217.4 |
| Gemini-2.5-pro[29] | 42.7 | 52.0 | 75.7 | 24.8 | 26.0 | 58.4 | 19.6 | 72.9 | | 1263.9 |
| *System-2 Models* | | | | | | | | | | |
| OpenAI-o1[77] | 49.5 | 48.7 | 54.2 | 48.9 | 36.1 | 55.3 | 59.1 | 43.0 | 55.1 | 4942.6 |
| *Human Performance* | | | | | | | | | | |
| Human | 77.5 | 88.2 | 70.9 | 90.2 | 77.2 | 87.4 | 71.2 | 78.5 | 69.2 | - |

evaluation is performed using the DeepSeek API[3], with customized prompts tailored to each question type to ensure accurate and consistent scoring. Experiments are conducted on NVIDIA A800 GPUs. Additional implementation details are provided in the Appendix G.

## 4.2 Experimental Results

We show the evaluation results of various models on SOLIDGEO in Table 3 and report the model performance by difficulty level, question type, and under different prompts in Table 4.

**Challenging Nature of SOLIDGEO.** The results presented in Table 2 highlight the inherent difficulty of SOLIDGEO. The best-performing model, OpenAI-o1, achieves an overall accuracy of 49.5%, followed by Gemini-2.5-pro at 42.7%, and Claude-3.7-Sonnet at 34.1%. Notably, all other models fall below the 30% threshold, underscoring the substantial challenge that solid geometry reasoning poses even for advanced MLLMs. Among the open-source system-1 models, the strongest model is Llama4, a 400B MoE model with 128 experts and 17B active parameters, which achieves 29.6% accuracy. In the System-2 category, the best-performing open-source model is QvQ-72B, reaching 26.6%. Despite these results, even the most advanced model OpenAI-o1 still falls significantly short of human performance, highlighting the limitations of current MLLMs in spatial reasoning. Additionally, there remains a notable performance gap between open-source and closed-source models, which shows the need for continued advancements in training strategies, data quality, and architectural innovations.

---

[3] https://www.deepseek.com/

Table 4: Model Performance across Different Prompt Settings, Difficulty Levels, and Question Types. MC: Multiple Choice, SS: Single-Step, MS: Multi-Step.

| Models | Original | CoT | CoT &2-shot | Level 1 | Level 2 | Level 3 | MC | SS | MS |
|---|---|---|---|---|---|---|---|---|---|
| Claude-3.5-sonnet [48] | **22.2** | 21.7(-0.5) | 19.3(-1.9) | 37.7 | 15.7 | 4.5 | 32.8 | 17.8 | 13.9 |
| Gemini-1.5-pro [29] | 25.3 | 26.1(+0.8) | **27.5**(+2.2) | 39.0 | 19.7 | 8.5 | 32.1 | 23.1 | 14.4 |
| GPT-4V [15] | **25.3** | 24.1(-1.2) | 23.4(-1.9) | 41.6 | 18.5 | 6.8 | 35.2 | 22.1 | 10.6 |
| GPT-4o [28] | **25.5** | 24.9(-0.6) | 22.9(-2.6) | 38.9 | 20.1 | 7.9 | 32.7 | 23.2 | 13.0 |
| Claude-3.7-Sonnet [48] | 34.1 | - | - | 42.5 | 31.2 | 16.4 | 39.9 | 32.5 | 21.2 |
| Gemini-2.5-pro [29] | 42.7 | - | - | 22.1 | **50.8** | **80.7** | 35.2 | 49.0 | 19.3 |
| OpenAI-o1 [77] | 49.6 | - | - | **46.6** | 50.4 | 57.4 | **50.8** | **51.2** | **28.8** |
| LLaVA-OneVision-Chat-7B [49] | 8.6 | 10.5(+1.9) | **11.4**(+2.8) | 16.1 | 5.4 | 1.1 | 22.1 | 2.6 | 1.4 |
| Qwen2.5-VL-Instruct-7B [30] | 15.5 | **16.0**(+0.5) | 15.0(-0.5) | 28.1 | 10.0 | 2.8 | 22.2 | 13.1 | 6.2 |
| LLaVA-OneVision-Chat-72B [49] | **15.9** | 14.8(-1.1) | 15.6(-0.3) | 23.5 | 13.0 | 4.0 | 29.2 | 10.2 | 6.2 |
| InternVL3-8B [16] | 17.7 | 18.1(+0.4) | **18.3**(+0.6) | 30.7 | 11.9 | 6.2 | 27.3 | 14.1 | 6.2 |
| Qwen2.5-VL-Instruct-72B [30] | 24.2 | **28.8**(+4.6) | 26.1(+1.9) | 33.5 | 20.6 | 10.6 | 28.5 | 22.8 | **17.3** |
| InternVL3-78B [16] | **26.2** | 25.8(-0.4) | 25.9(-0.3) | 42.2 | 19.7 | 6.2 | 35.1 | 23.0 | 15.4 |
| QvQ-72B-Preview [76] | 26.6 | - | - | 32.6 | 24.5 | 14.7 | 31.5 | 26.3 | 6.7 |

**Comparison among different Subjects.** As shown in Table 2, models generally underperform on tasks requiring complex spatial reasoning, such as Planar Unfolding and Configuration (PUC), Multi-view Projection (MVP), 3D Coordinate and Vector Reasoning (3DCV). Even the strongest model, OpenAI-o1, achieves only 36.1% on PUC and 43.0% on MVP. Interestingly, Gemini-2.5-pro exhibits a notable performance pattern—surpassing human-level accuracy on 3DCV and Spatial Metric Relations (SMR), the most cognitively demanding categories for humans, while underperforming on comparatively simpler tasks where humans excel, such as Solid Shape Identification (SSI).

**Failure of CoT/few-shot.** As shown in Table 4, Chain-of-Thought (CoT) and few-shot prompting do not consistently improve model performance on SOLIDGEO. Models like Qwen2.5-VL-72B and Gemini-1.5-pro benefit from CoT and 2-shot settings, with accuracy increasing from 24.2% to 28.8% and 26.1%, and from 25.3% to 26.1% and 27.5%, respectively. However, other models such as GPT-4o and LLaVA-OV-72B exhibit performance degradation under the same conditions. To further understand this, we analyze the effect of prompting strategies on the average number of generated tokens, as shown in Figure 4. For many models, neither CoT nor 2-shot leads to longer outputs, suggesting such reasoning capabilities may already be learned during pretraining. Notably, we

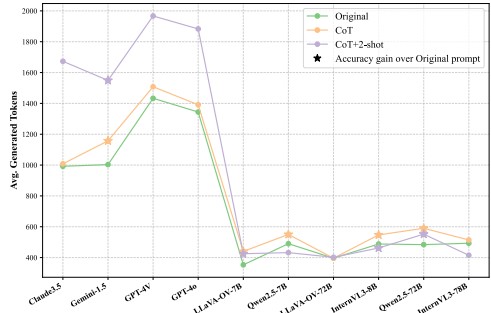

Figure 4: Average number of generated tokens for different prompting strategies across various models on SOLIDGEO.

observe that only when CoT increases token count by over 10% does it yield tangible gains—e.g., Qwen2.5-VL-72B: tokens ↑21.9%, accuracy ↑4.6%; LLaVA-OV-7B: tokens ↑20.6%, accuracy ↑1.9%. In contrast, 2-shot prompting shows no consistent correlation with performance improvement, indicating it is not a universally effective strategy. Efficient long-context modeling remains a challenge, despite recent advances in scalable context generalization techniques [79].

**Difficulty Levels and Question Types.** Analyses in this paragraph are conducted under the original prompt setting. As shown in Table 4, most models exhibit a clear decline in accuracy as task difficulty increases. For instance, InternVL3-78B achieves 42.2% on Level 1 but drops sharply to just 6.2% on Level 3, underscoring the increasing complexity of harder solid geometry problems. Interestingly, Gemini-2.5-pro and OpenAI-o1 defy this trend—their performance improves with higher difficulty levels. Gemini-2.5-pro reaches an impressive 80.7% on Level 3 while scoring only 22.1% on Level 1, suggesting a potential overfitting to complex problem structures or a struggle with generalizing across simpler formats. Regarding question types, most models perform better on multiple-choice (MC) questions, likely due to reduced ambiguity from predefined options. However, Gemini-2.5-pro and OpenAI-o1 once again diverge from this pattern, achieving their highest scores on single-step (SS) questions. This counterintuitive behavior may reflect an internal ability to dynamically adjust reasoning strategies based on task complexity.

**Model Inference Efficiency.** As shown in Figure 5, System-2 models generally generate much longer outputs than System-1 models. While this often improves accuracy, it reduces inference

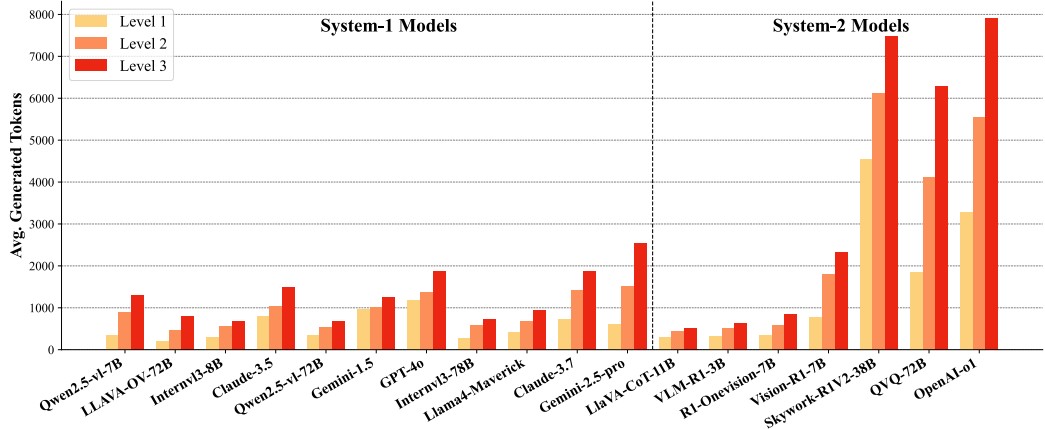

Figure 5: Average number of generated tokens by difficulty level of various models on SOLIDGEO.

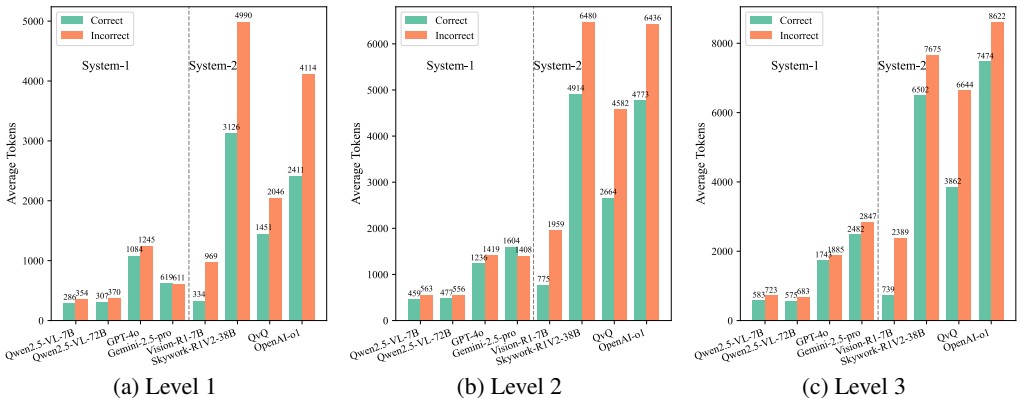

| (a) Level 1 | (b) Level 2 | (c) Level 3 |

Figure 6: Average generateed tokens by correctness across different complexity levels.

efficiency due to longer reasoning and latency. We also observe that output token counts increase with problem difficulty, suggesting that models require longer reasoning chains for more complex tasks. To better understand the link between output length and correctness, we analyze token usage by answer correctness across difficulty levels in Figure 6. Results show that models—especially System-2—tend to consume more tokens for incorrect answers than correct ones, without accuracy gains. This implies potential overthinking, where excessive reasoning fails to improve outcomes. Reducing unnecessary reasoning steps while maintaining performance remains a key challenge for building efficient and capable MLLMs. We provide further analysis in Appendix E.

## 4.3 Error Analysis

We conduct a fine-grained error analysis on two representative models: Gemini-2.5-pro and OpenAI-o1. Specifically, we randomly sample 300 incorrect predictions from each model, classify the errors into five categories, and analyze their distribution. The error taxonomy and proportions are visualized in Figure 7, with more details provided in the Appendix H.

Among all error types, visual perception errors and reasoning errors dominate, jointly accounting for over 70% of total failures. OpenAI-o1 exhibits a lower reasoning error rate (38%) compared to Gemini-2.5-pro (44%), suggesting that System-2 models with long CoT reasoning paradigms may partially mitigate faulty logical inference. However, OpenAI-o1 suffers from a higher visual perception error rate (41% vs. 34%), indicating that even advanced models struggle to accurately interpret spatial geometric elements. Moreover, we observe that OpenAI-o1 produces hallucinated content in 6% of cases—double the rate observed in Gemini-2.5-pro. This highlights hallucination as an emerging issue in complex long CoT generation settings, especially for System-2 models. An illustrative reasoning error is shown on the right side of Figure 7, where the model incorrectly applies logical reasoning without accounting for spatial structure.

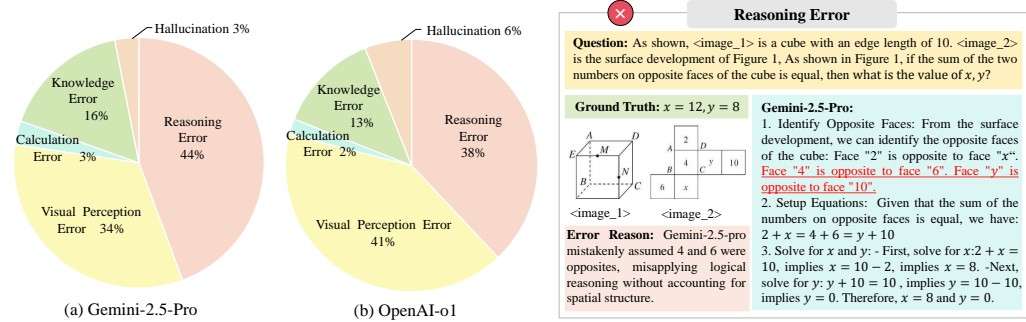

Figure 7: Error distribution of Gemini-2.5-pro and OpenAI-o1 and an example of reasoning error.

## 5 Conclusion

We introduce SOLIDGEO, the first large-scale benchmark designed to evaluate the solid geometry mathematical reasoning capabilities of MLLMs, addressing a critical gap in current multimodal benchmarks that overlook spatial and three-dimensional mathematical reasoning tasks. SOLIDGEO comprises a diverse and challenging set of real-world K–12 and competition-level problems enriched with visual contexts, difficulty levels, and fine-grained categorical annotations. Using this benchmark, we conduct a comprehensive evaluation of a wide range of open-source and closed-source models, revealing a significant performance gap between current MLLMs and human performance on solid geometry tasks. Furthermore, we analyze model inference efficiency across difficulty levels and response lengths, offering valuable insights into the current limitations and potential directions for future research. We hope SOLIDGEO will serve as a foundation for advancing the complex spatial reasoning capabilities of next-generation MLLMs.

**Limitations and Future Works.** SOLIDGEO has certain limitations. First, the dataset includes only English and Chinese problems, which may limit its applicability for evaluating multilingual reasoning capabilities [80]. Second, there currently exists no widely adopted formal language to represent solid geometry problems, making it difficult to standardize symbolic reasoning in this domain. Future work could explore designing a formal representation framework for solid geometry, which may further facilitate precise modeling and programmatic evaluation.

## 6 Acknowledgements

This work has been supported by the National Natural Science Foundation of China (NSFC) Grant 62436009.

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

# Appendix

## A More Detailed Statistics about SOLIDGEO

In this chapter, we will introduce more statistics about our SOLIDGEO. As shown in 2, every question in SOLIDGEO contains at least one image input, annotated with difficulty levels and fine-grained solid geometry categories.

### A.1 Distribution of Text Length

Questions in SOLIDGEO are presented in English or Chinese. As shown in Table 1, the longest question in SOLIDGEO spans 679 words, with an average length of 77.2 words. Figure 12 further illustrates the distribution of text lengths, highlighting the diversity of SOLIDGEO. The length of English and Chinese questions is counted in words and Chinese characters, respectively.

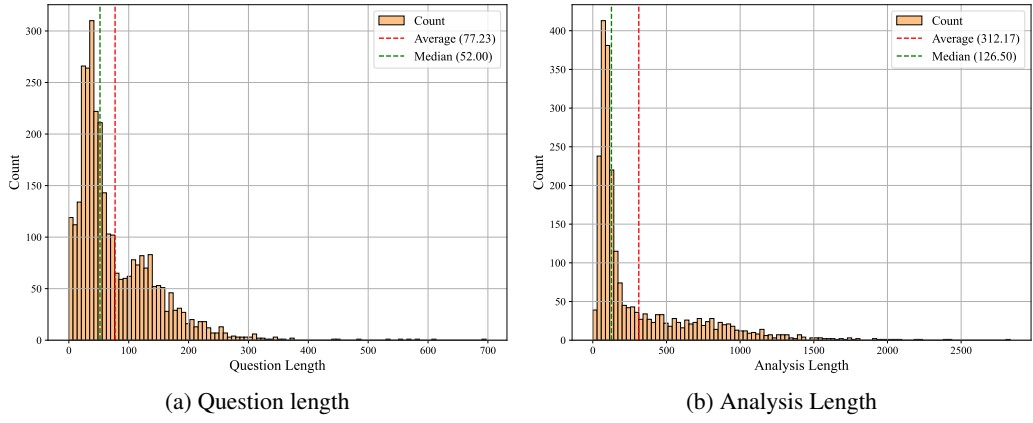

(a) Question length  (b) Analysis Length

Figure 8: The distribution of the length per question in SOLIDGEO.

### A.2 Distribution of Image Number per question

As shown in Figure 9, the majority of questions in the SOLIDGEO dataset (74.5%) are accompanied by a single image. The remaining questions are associated with multiple images, with the number of images ranging from two to eight.

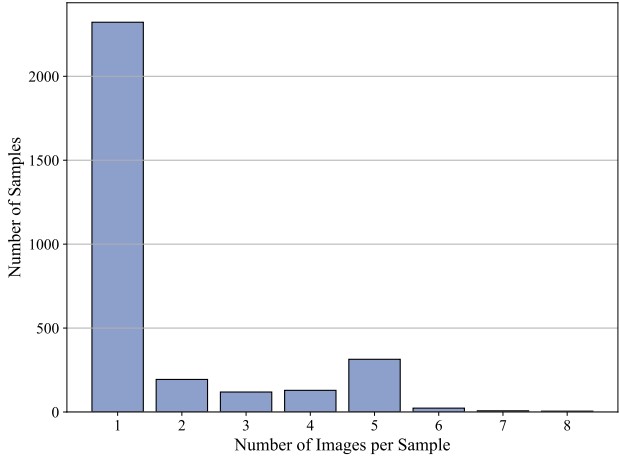

Figure 9: The distribution of the number of images per question in SOLIDGEO.

## A.3 Distribution of Solid Geometry Subjects

Figure 10 presents the distribution of subject categories within the solid geometry portion of the SOLIDGEO. The questions are categorized into eight distinct geometry-related topics. Among them, the most dominant subject is *Spatial Metric Relations*, accounting for 857 samples, followed by *Composite Solid Structures* (794) and *Multi-view Projection* (699).

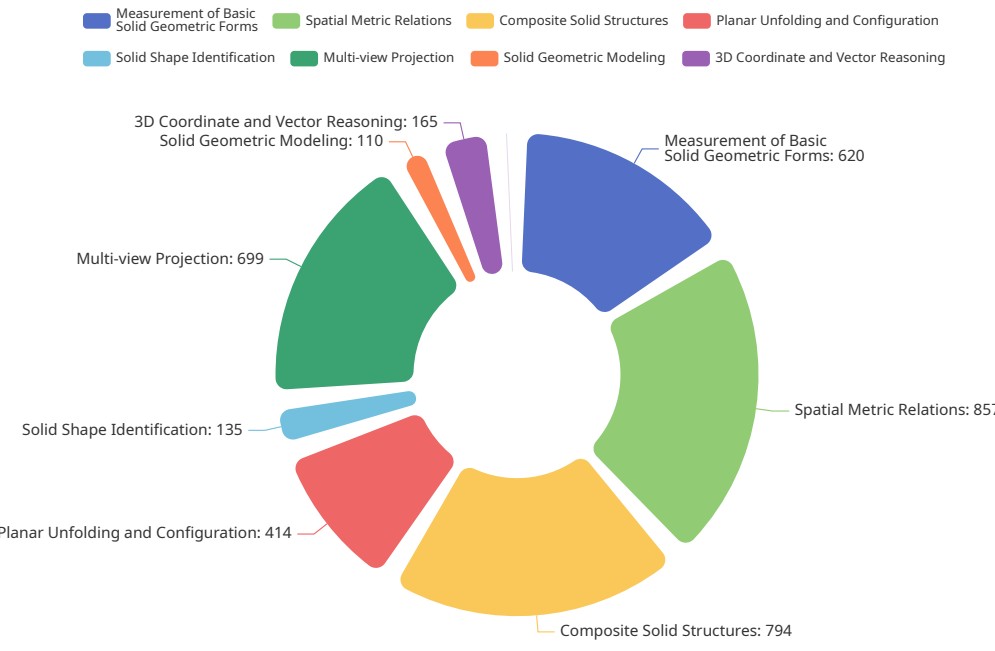

Figure 10: The distribution of 8 fine-grained solid geometry subjects in SOLIDGEO.

*Measurement of Basic Solid Geometric Forms* also comprises a substantial portion with 620 samples. Mid-sized categories include *Planar Unfolding and Configuration* (414) and *3D Coordinate and Vector Reasoning* (165). Notably, even the smallest categories—*Solid Shape Identification* (135) and *Solid Geometric Modeling* (110)—still contain more than 100 samples each, which is sufficient to ensure evaluation reliability and statistical stability across all sub-domains.

This distribution demonstrates that while the dataset exhibits a certain degree of imbalance, all categories are adequately represented to support meaningful performance comparisons for models across different solid geometry skills.

## A.4 Distribution of Sources

Figure 11 illustrates the composition of our dataset in terms of source origin. The dataset integrates solid geometry problems from seven distinct benchmarks, including MathVerse (11.47%), MathVision (7.84%), GeoEval (2.41%), DynaMath (4.49%), OlympiadBench (24.93%), and CMMaTH (4.66%). Notably, 44.2% of the samples are newly collected and curated solid geometry problems, which significantly expand the diversity and coverage of our dataset.

To ensure the quality and uniqueness of each problem, we performed rigorous filtering and deduplication across all sources. Redundant or low-quality items were removed, and only samples with well-defined geometric settings, unambiguous language, and reliable answer annotations were retained. This refinement process guarantees that each question in the final SolidGeo dataset is both high-quality and non-overlapping, providing a consistent and trustworthy benchmark for evaluating spatial reasoning capabilities. For the specific process, see Appendix D.

This broad integration of sources—combined with strict curation—ensures that the dataset covers a wide range of solid geometry problem types while maintaining a high standard of clarity, correctness, and diversity.

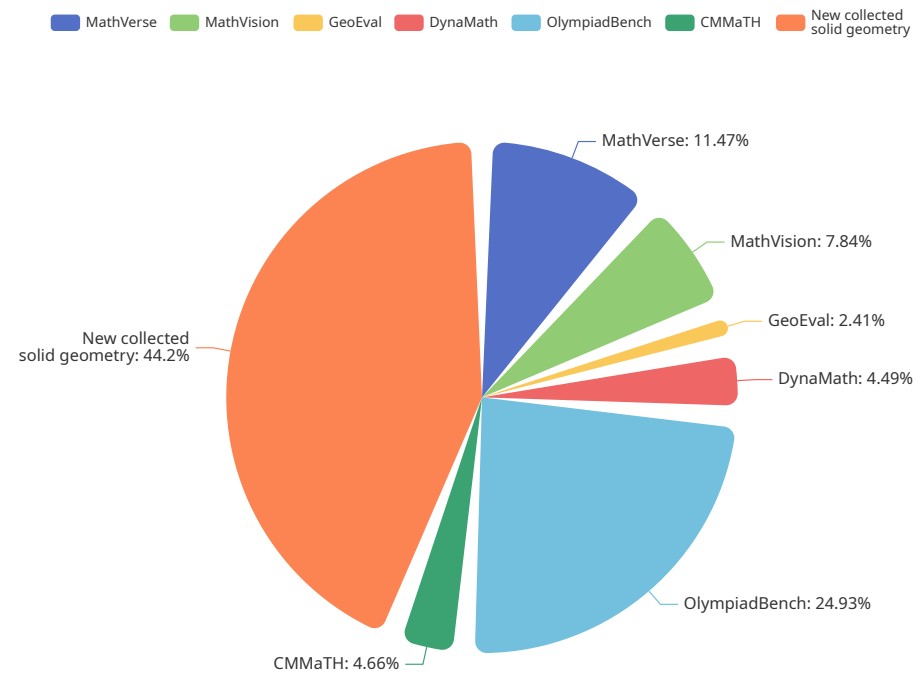

Figure 11: The distribution of different sources in SOLIDGEO.

## B  Introduction of our Fine-grained Solid Geometry Subjects

In this section, we introduce the eight fine-grained subject categories defined in our solid geometry benchmark. These categories are designed to reflect the diverse reasoning skills required for real-world spatial understanding tasks. Rather than treating solid geometry as a monolithic domain, we present a structured taxonomy that decomposes the problem space into conceptually distinct yet complementary components. Figure 22 to Figure 29 illustrate representative image examples corresponding to each category.

Our classification is based on both the cognitive processes involved (e.g., spatial metric reasoning, 3D transformation) and the structural features of the geometric objects (e.g., solid composition, projection, unfolding). This allows for more interpretable performance diagnostics and supports targeted model evaluation. The eight categories span a broad range of solid geometry competencies, ensuring coverage of foundational as well as advanced visual-spatial skills.

**1. Measurement of Solid Geometric Forms.** This category focuses on fundamental formula-based computations involving standard three-dimensional shapes such as cubes, cuboids, cylinders, cones, spheres, and regular polyhedrons. Problems typically involve direct application of geometric formulas for volume, surface area, or edge length based on given dimensions. These problems often serve as entry points in spatial reasoning tasks and require little to no shape manipulation or visualization beyond understanding the formula's variables. *Example task:* Compute the volume of a right circular cone with radius 4 cm and height 9 cm.

**2. Solid Shape Identification.** This subject class targets the recognition and naming of 3D geometric solids or their components (such as faces, edges, or vertices), based on visual or structural cues. The focus lies in spatial visualization rather than computation. Problems typically include diagrams and ask students to match shapes to names or count features (e.g., number of faces). *Example task:* Identify the name of a 3D shape with 8 faces and 12 edges, or label the edges in a net diagram.

**3. Spatial Metric Relations.** This category involves reasoning about geometric measurements in 3D space, particularly those that rely on theorems and spatial relationships such as distances, angles, and relative positions. Solutions typically require applying properties of perpendicularity, intersection, or symmetry. These problems demand an understanding of spatial configurations and their mathematical implications. *Example task:* Determine the angle between a diagonal of a cube and one of its faces, or calculate the shortest distance from a point to a plane.

**4. Multi-view Projection.** This class includes problems involving orthographic projections and their interpretation. It emphasizes the ability to switch between 2D projections (front, top, side views) and 3D spatial understanding. Learners must mentally reconstruct 3D solids from multiple projections or generate accurate views based on a given model. *Example task:* Infer the 3D object described by its top and front views, or draw the side view of a given solid.

**5. Planar Unfolding and Configuration.** This subject covers the analysis of how 3D solids unfold into 2D nets and vice versa. It includes determining valid unfoldings, reasoning about face connectivity, and reconstructing solid forms from nets. It also addresses spatial folding logic and pathfinding over surfaces (e.g., shortest path problems on a cube's surface). *Example task:* Complete the missing face in the net of a cube, or determine if a given net corresponds to a regular dodecahedron.

**6. Composite Solid Structure.** Problems in this category deal with complex solids formed by combining, intersecting, or modifying standard geometric shapes. It requires understanding how operations such as union, subtraction, and intersection affect volume and surface area. This type reflects realistic modeling scenarios involving multi-body systems. *Example task:* Find the volume of the intersection between a sphere and a cube, or compute the surface area of a cylinder with a conical cutout.

**7. 3D Coordinate and Vector Reasoning.** This subject category utilizes algebraic methods to solve geometric problems in three-dimensional coordinate systems. It includes vector-based calculations of angles, distances, projections, and normal vectors. Problems often translate geometric relationships into equations and exploit coordinate geometry principles. *Example task:* Calculate the angle between two vectors or find the shortest distance from a point to a line in space.

**8. Solid Geometric Modeling.** This is the most application-oriented category, featuring problems that simulate real-world use of solid geometry in architecture, engineering, and design. It involves optimizing parameters, validating geometric constraints, or modeling complex surfaces. This class bridges pure geometry with practical problem-solving. *Example task:* Design an optimized water tank shape with minimal surface area or verify whether a given structure satisfies balance and symmetry constraints.

## C   Comparison

As shown in Section 3.3, we have already compared SOLIDGEO with existing datasets in terms of coverage and task diversity. In this section, we further extend the comparison from two additional perspectives: (1) the average length of questions, and (2) the performance of the best-performing model on each dataset.

As shown in Figure 12, our SOLIDGEO exhibits the longest average question length among all compared benchmarks, reaching 77.2 words. This is significantly higher than datasets such as MathVista (15.6), GeoQA (37.1), and MMMU-MATH (40.8). The increased question length in SOLIDGEO directly reflects the need for more precise problem understanding and more complex reasoning processes, which distinguishes it from benchmarks that primarily involve shorter and more formulaic tasks. Moreover, the right subfigure highlights a striking performance gap: despite the state-of-the-art models achieving over 80% accuracy on other benchmarks, their best performance on SOLIDGEO drops sharply to 49.5%. This further validates the higher reasoning difficulty and challenge posed by our benchmark. It underscores the necessity of more robust spatial understanding and compositional reasoning to succeed on SOLIDGEO.

## D   More Detailed Construction of SOLIDGEO

To further clarify the construction process of SOLIDGEO, this section provides additional details on the data collection strategy, filtering pipeline, and annotation workflow introduced in Section 3. These

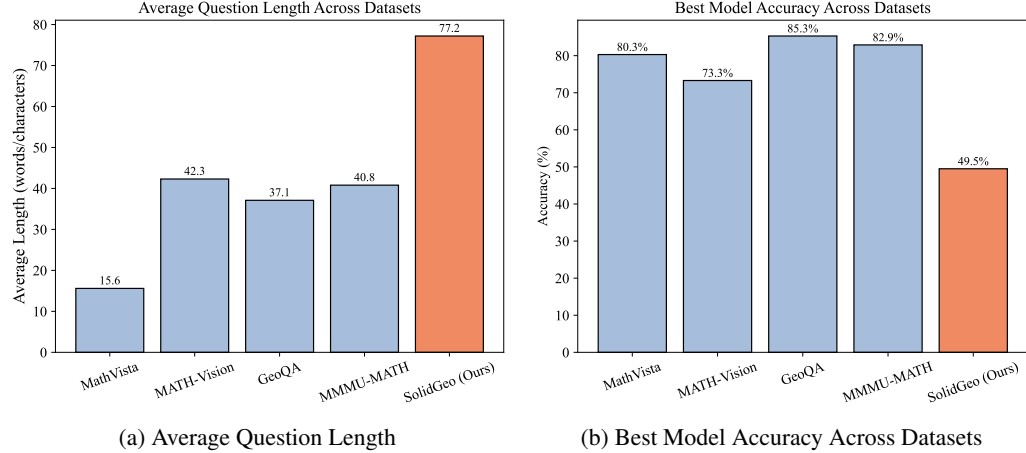

(a) Average Question Length                    (b) Best Model Accuracy Across Datasets

Figure 12: Average question length and best-performing model accuracy across datasets

components jointly ensure that each problem in SOLIDGEO is high-quality, topically relevant, and uniquely represented. We first describe the hybrid data sourcing approach, followed by our four-stage filtering procedure and the multi-phase annotation protocol used to assign difficulty levels and subject categories.

## D.1 Data Collection Details

To construct SOLIDGEO, we adopted a hybrid data collection strategy that combines both existing benchmarks and newly collected real-world data. Specifically, we extracted solid geometry problems from six publicly available multimodal math datasets: **MathVerse** [7], **MATH-Vision** [6], **Dyna-Math** [63], **GeoEval** [5], **CMMaTH** [81], and **OlympiadBench** [64], based on their official topic annotations or keyword filtering. These samples account for 1,737 problems in our final dataset.

To enrich the dataset with more diverse and realistic problem styles, we further collected **1,376 new solid geometry problems** from K–12 education sources. Most of these were obtained from the Zujuan platform[4], a widely used educational resource that provides large-scale math problems in PDF format. We employed the Mathpix OCR API [5] to extract structured problem components, including question texts, diagrams, and answer analyses.

The initial candidate pool contained 10,932 problems after solid geometry keyword filtering, which were later filtered and refined through a four-stage pipeline (see Section 3.2). These newly collected samples contribute 44.2% to the final dataset, significantly enhancing its diversity in language, format, and reasoning complexity.

## D.2 Data Filtering Details

**Recap of the Four-Stage Filtering.** As described in Section 3.2, we employed a systematic four-stage filtering pipeline: (1) structural filtering to ensure multimodal completeness, (2) image quality filtering using OpenCV, (3) semantic filtering using large model-based topic classification, and (4) n-gram-based deduplication across datasets.

**Image Quality Filtering with OpenCV.** To remove blurry or low-resolution diagrams, we computed the variance of the Laplacian for each image as a sharpness metric. A conservative threshold of **1000** was adopted; images with a variance below this threshold were excluded from the dataset. This ensured that all retained images contain sufficiently clear visual information to support accurate spatial reasoning.

**Cross-Set Deduplication via Post-hoc n-gram Similarity Analysis.** Instead of applying rule-based filtering, we conducted a comprehensive word-level $n$-gram similarity analysis between our newly

---

[4]`https://zujuan.xkw.com/`

[5]`https://mathpix.com/convert`

collected `solidgeo` problems and existing benchmark samples. Specifically, we computed the cosine similarity between each `solidgeo` question and all external questions based on their $n$-gram count vectors, with $n = 2, 3, 4, 5, 8$.

Let $S_{i,j}$ denote the cosine similarity computed between the word-level $n$-gram feature vectors of the $i$-th `solidgeo` question and the $j$-th `existing` question. The average cross-source similarity is then calculated as:

$$\text{Average Similarity} = \frac{1}{N_{\texttt{solidgeo}} \times N_{\texttt{existing}}} \sum_{i=1}^{N_{\texttt{solidgeo}}} \sum_{j=1}^{N_{\texttt{existing}}} S_{i,j}$$

We observed that the average similarity drops rapidly as $n$ increases—e.g., from $2.53\%$ at $n = 2$ to near-zero at $n = 8$, validating the lexical independence of SOLIDGEO. This fully demonstrates that there is no overlap between our newly collected `solidgeo` and existing samples. The detailed statistics of word-level $n$-gram similarity between SolidGeo and existing sources are summarized in Table 5.

Table 5: Word-level $n$-gram Average Similarity Between `SolidGeo` and Other Sources

| Source Pair | 2-grams | 3-grams | 4-grams | 5-grams | 8-grams |
|---|---|---|---|---|---|
| SolidGeo vs Others | 2.53% | 0.60% | 0.29% | 0.10% | 0.00% |

### D.3 Data Annotation Protocol

**Difficulty Labeling via Prompted Model Voting.** To label question difficulty, we constructed a tailored prompt based on expert-provided heuristics. This prompt guides LLMs to judge the complexity of geometric reasoning required (e.g., direct formula use vs. multi-step deduction). Three advanced MLLMs—**GPT-4o**, **Claude-3.7-Sonnet**, and **Qwen-VL-Max**—were queried in parallel, and a majority vote was used to assign a difficulty level from 1 (easy) to 3 (hard). For cases where model votes disagreed, the final label was manually assigned by experienced annotators. Detailed prompt for difficulty level is shown in Table 6.

**Subject Categorization Based on Expert Taxonomy.** To define meaningful subject categories, we invited two domain experts in geometry education to review a representative subset of our data. Based on their analysis of geometry curricula and problem-solving cognitive patterns, they proposed an 8-category taxonomy: *Measurement of Solid Geometric Forms*, *Solid Shape Identification*, *Spatial Metric Relations*, *Multi-view Projection*, *Planar Unfolding and Configuration*, *Composite Solid Structure*, *3D Coordinate and Vector Reasoning* and *Solid Geometric Modeling*. These categories aim to reflect essential dimensions of solid geometry reasoning, ensuring interpretability and task-specific performance analysis. See Table 6 for detailed prompt.

Given the taxonomy, the same three MLLMs (GPT-4o, Claude-3.7-Sonnet, and Qwen-VL-Max) were prompted to predict the subject of each problem. Again, majority voting was used to assign a label, and human annotators resolved any disagreements.

**Annotator Review via Interactive Annotation Platform.** After automated labeling, we developed an internal web-based interface to visualize problem texts, diagrams, and model predictions. Using this tool, three trained annotators independently reviewed all 3,113 samples. Each problem was checked for both subject and difficulty correctness. In cases of disagreement, consensus was reached through discussion. This final human verification stage ensures label consistency and semantic integrity across the dataset.

## E Generated Token Distribution

Figures 13–16 show the token distributions of System-1 models, while Figures 17–21 correspond to System-2 models. In general, System-2 models tend to produce longer outputs. Across most models, the generated token distributions exhibit long-tailed characteristics.

Notably, **Gemini-2.5-pro** and **OpenAI-o1** demonstrate the most evident long-tail distributions, suggesting their ability to dynamically adjust output length based on problem complexity. Such adaptive behavior is crucial for efficient reasoning and future inference scalability.

In contrast, models like **Skywork-R1V2-38B** and **Vision-R1-7B** show highly concentrated output lengths, indicating a lack of dynamic adjustment in response to varying problem difficulty—i.e., these models tend to *overthinking* [82] even for simpler questions.

Interestingly, although **R1-Onevision-7B** is a System-2 model, its token distribution is relatively narrow. This may be due to model size limitations that prevent the expression of extended multi-step reasoning, indicating possible *underthinking* [83] behavior.

Overall, both overthinking and underthinking behaviors are emerging concerns in current models. Developing mechanisms to dynamically adapt reasoning depth based on problem complexity is essential for achieving robust and efficient multimodal reasoning.

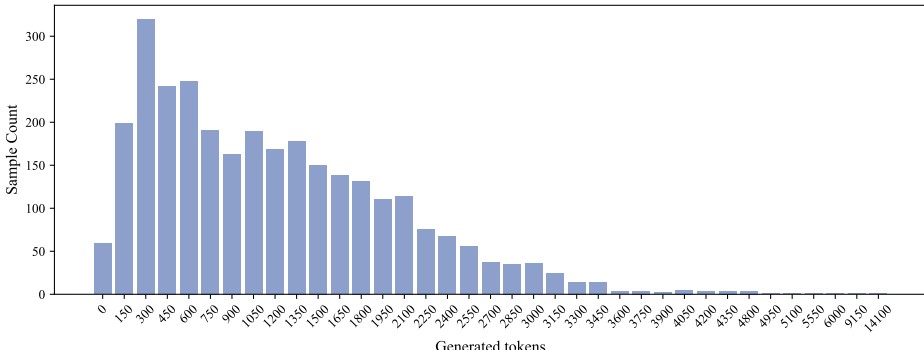

Figure 13: The generated token distribution of Gemini-2.5-pro on SOLIDGEO.

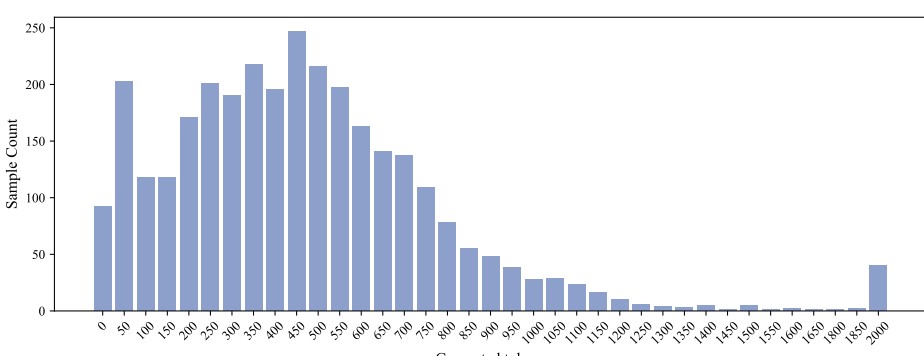

Figure 14: The generated token distribution of Internvl3-78B on SOLIDGEO.

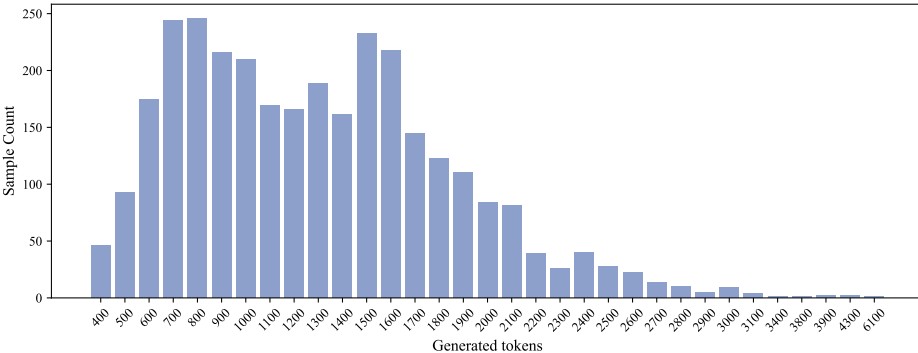

Figure 15: The generated token distribution of GPT-4o on SOLIDGEO.

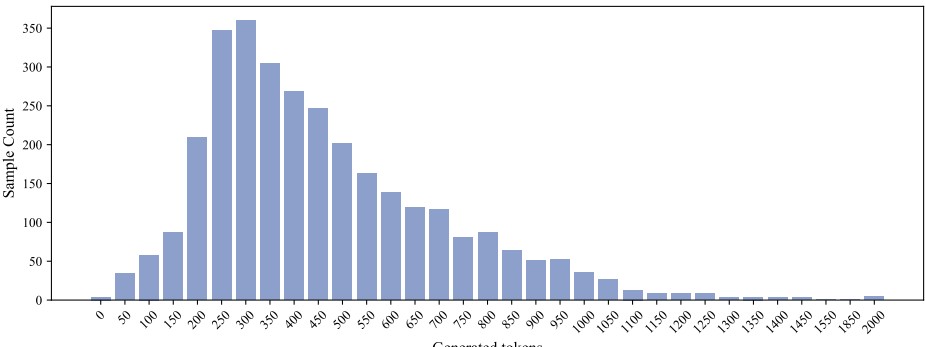

Figure 16: The generated token distribution of Qwen2.5-VL-72B on SOLIDGEO.

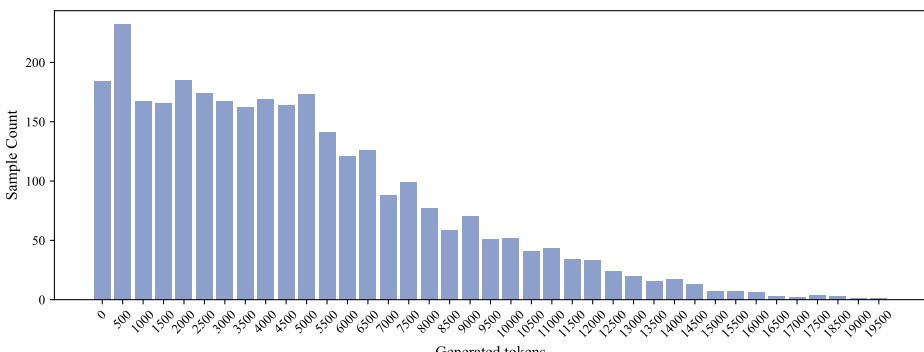

Figure 17: The generated token distribution of OpenAI-o1 on SOLIDGEO.

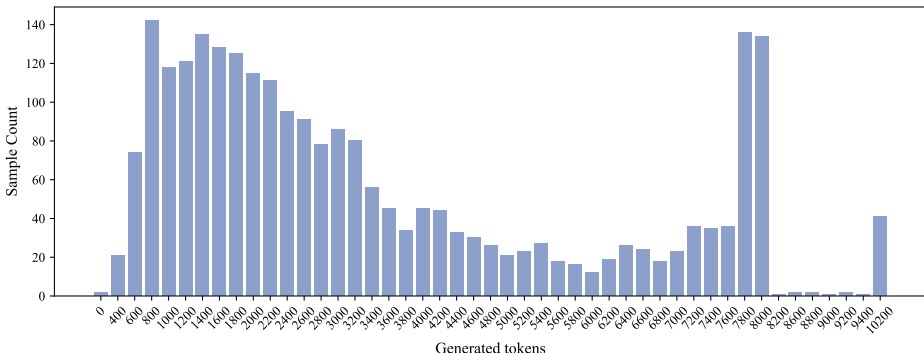

Figure 18: The generated token distribution of QvQ-72B on SOLIDGEO.

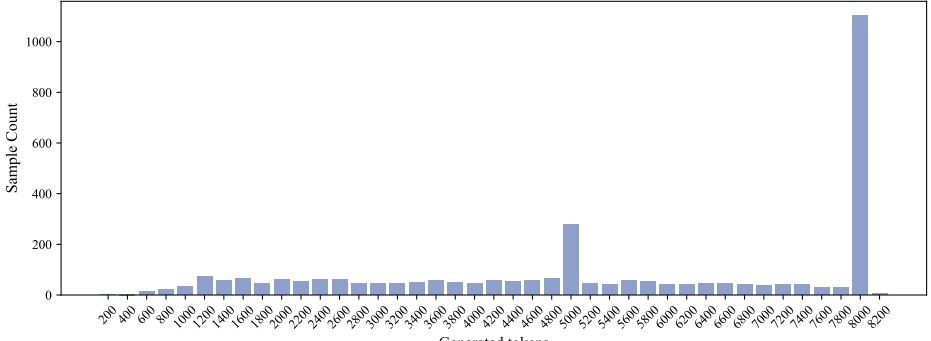

Figure 19: The generated token distribution of Skywork-R1V2-38B on SOLIDGEO.

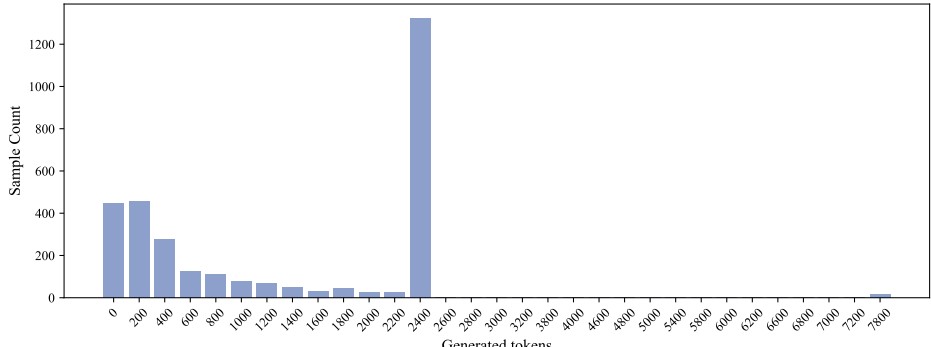

Figure 20: The generated token distribution of Vision-R1-7B on SOLIDGEO.

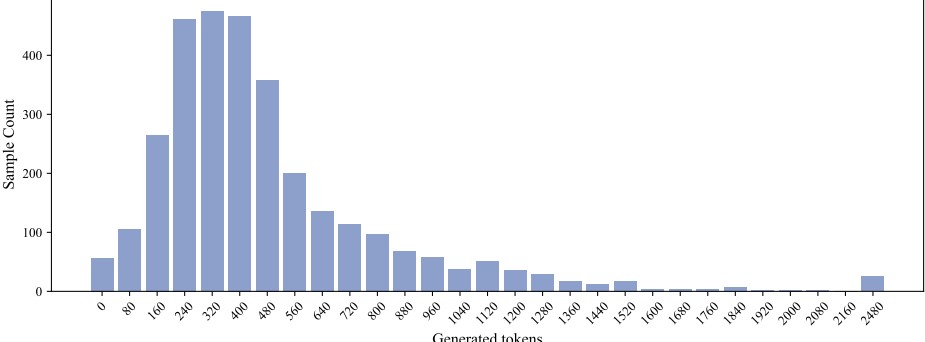

Figure 21: The generated token distribution of R1-Onevision-7B on SOLIDGEO.

Table 6: Prompt Design for Problem Classification and Annotation

**System Prompt**

You are a 3D geometry problem classification expert. Your task is to analyze solid geometry problems and output structured annotations. You should extract geometric elements, spatial relationships, calculation targets, classify the problem type, and assign a reasoning complexity level. Respond in JSON format, and do not include explanations.

**User Prompt for Difficulty Level Classification**

Please assign a difficulty level (1: easy, 2: medium, 3: hard) to the following solid geometry problem. Use the following criteria:
- Level 1: Basic Execution — Problems in this category require the direct application of a single formula or a well-known geometric rule. The solution path is straightforward with minimal intermediate steps.
- Level 2: Multi-Step Reasoning — Problems involve several computational steps or the construction of simple proofs. They require organizing multiple procedures in a logical sequence and understanding intermediate relationships between geometric elements.
- Level 3: Creative Spatial Reasoning — Problems in this level necessitate advanced spatial visualization and innovative approaches. Instead of a standard formula, solvers must explore geometric relationships and often create non-obvious solution strategies.
Return only the level number.

**User Prompt for Category Classification**

You are a geometry expert. Given a solid geometry problem, classify it into one or more of the following categories based on its core reasoning focus:
1. Measurement of Basic Solid Geometric Forms. Key Features: Formula-driven computations for regular polyhedrons and revolution solids (e.g., Platonic solids, cylinders, cones, spheres). *Example: "Calculate the volume of a regular octahedron with 3cm edge length".*
2. Solid Shape Identification. Key Features: Visual recognition of geometric solids or their components (faces/edges/vertices) without quantitative analysis. *Example: "Identify the name of the 3D shape shown in the diagram".*
3. Spatial Metric Relations. Key Features: Calculations involving spatial relationships (point-to-plane distances, line-to-surface angles) using geometric theorems. *Example: "Determine the shortest distance from a point to a given plane in 3D space".*
4. Multi-view Projection Analysis. Key Features: Conversion between 2D orthographic projections (front/top/side views) and 3D reconstructions. *Example: "Reconstruct a solid model from its front and top views".*
5. Planar Unfolding and 3D Configuration. Key Features: Analysis of surface unfoldings (nets) and spatial folding patterns, including path optimization. *Example: "Complete the missing face in a dodecahedron's net".*
6. Composite Solid Structural Analysis. Key Features: Solving problems involving intersecting/cut/combined solids (e.g., Boolean operations on shapes. *Example: "Find the intersection volume of a sphere and a cube".*
7. 3D Coordinate and Vector Applications. Key Features: Algebraic solutions using 3D coordinate systems, vectors, or parametric equations. *Example: "Compute the angle between two spatial vectors"* .
8. Engineering Solid Geometric Modeling. Key Features: Real-world geometric optimization and structural validation (e.g., architectural/mechanical design). *Example: "Optimize the curved surface design of a water storage tank".*
Output the predicted categories as a list. Use multiple labels if necessary.

# F   Annotator and Human Level

To ensure high-quality and reliable annotations, we assembled a team comprising two domain experts and several trained annotators. The two experts possess extensive experience in the field of solid geometry and played a key role in designing the annotation schema, defining the eight fine-grained subject categories, and establishing the difficulty level criteria.

All annotators are graduate students with strong mathematical backgrounds and spatial reasoning ability. They received targeted training and annotated the data strictly following expert-defined protocols. After the model-assisted initial annotation stage, each sample was independently reviewed by three annotators to verify the assigned subject category and difficulty level. This rigorous process ensures the consistency and credibility of all annotations used in our benchmark. Specifically, the expert-provided guidelines instructed annotators to classify each problem based on its core reasoning focus into one or more of the eight fine-grained solid geometry categories, such as spatial metrics, projection transformations, or coordinate modeling, and to assign a difficulty level according to a three-tier rubric: Level 1 for direct application of geometric formulas, Level 2 for multi-step deduction involving intermediate reasoning, and Level 3 for complex spatial reasoning or geometric abstraction. Annotators were asked to ignore irrelevant linguistic complexity, focus on reasoning-relevant elements, and flag ambiguous cases for expert arbitration to ensure labeling quality and inter-annotator consistency.

To establish a human performance baseline, we recruited 60 high school students and asked them to complete the questions independently, with each student assigned their own section of the questions without external assistance. Their responses were evaluated using the same criteria as the model. The results provide meaningful references for understanding the relative strengths and weaknesses of current multimodal models in solid geometry reasoning tasks.

We confirm that our human baseline study involving high school students was conducted under a formal and responsible ethical protocol. The study adhered to the following procedures:

- Informed Consent: Consent was obtained from the legal guardians of all student participants, alongside assent from the students themselves.

- Voluntary Participation: Participation was entirely voluntary, and students retained the right to withdraw at any time without penalty.

- Privacy Protection: All collected data was fully anonymized; no personal information was recorded or retained.

- Compensation: Participating students received a set of school-approved mathematics learning materials and educational tools. These items were selected to support students' academic growth and were appropriate for their grade level.

# G   Evaluation Details

For open-source models, all experiments—except for LLaMA-4-Maverick-17B-128E, which is accessed via API—are conducted locally on A800 GPUs. This setup ensures consistent hardware conditions for evaluating most models.

## G.1   Prompt for Response Generation

To ensure the model provides accurate responses, we design distinct CoT and 2-shot prompts tailored for multiple-choice, single-step, and multi-step free-form questions. The original prompt directly instructs the model to generate the final answer without intermediate reasoning. Detailed information can be found in Table 7, Table 8, Table 9.

## G.2   Prompt for Answer Evaluation

Our evaluation is conducted using the Deepseek API. For the evaluation of multiple-choice, single-step, and multi-step free-form questions, different prompts are designed to ensure accuracy in answer extraction and assessment. We first use the Deepseek API to extract the model's answers and then it

compares the extracted answers with the ground truth to determine the correctness of the answers. The specific prompts are shown in 10 below.

### G.3   Model Details

All experiments are conducted using models configured with a temperature of 0.2 for text generation. For System-1 models, we set the max_tokens limit to 4096, while for System-2 models—designed for more complex reasoning tasks—the limit is extended to 16,384. Comprehensive details regarding the models utilized in the evaluation are presented in Table 11.

| **Original** |
| --- |
| You are an assistant for solving math problems. Your input consists of a math question and images, give your answer directly, without any intermediate steps. |

| **CoT (Chain of Thought)** |
| --- |
| You are an assistant for solving math problems. Your input consists of a math question and images. Your task is to output the solution steps and the answer. The output format should be a step-by-step approach. Each question is multiple choice with one correct answer. Your final answer must be one of A, B, C, or D, and it should be placed within { }. For example: {A}, {B}, {C}, or {D}. |

| **CoT with 2-shot** |
| --- |
| Example 1: 
 Question: If a triangle has two sides of length 3 and 4, what is the length of the hypotenuse? 
 A.10 B.8 C.5 D.4 
 Answer: 
 Step 1 (Mathematical theorem used: Pythagorean theorem): The Pythagorean theorem states that in a right triangle, the square of the hypotenuse is equal to the sum of the squares of the other two sides. The formula is: $c^2 = a^2 + b^2$, where $a$ and $b$ are the legs, and $c$ is the hypotenuse. 
 Step 2 (Substitute the known values): Given $a = 3$ and $b = 4$. Substituting these values into the formula: $c^2 = 3^2 + 4^2 = 9 + 16 = 25$ 
 Step 3 (Calculate the hypotenuse): Taking the square root gives: $c = \sqrt{25} = 5$ 
 Answer: {C} 
 Example 2: 
 Question: In the right triangle ABC, AB is perpendicular to BC. It is known that AC=5 and AB=4. Find the area of the right triangle. A.20 B.10 C.5 D.6 
 Answer: 
 Step 1 (Mathematical theorem used: Pythagorean theorem): We first use the Pythagorean theorem to find the length of $BC$. The formula is: $AC^2 = AB^2 + BC^2$, where $AC$ is the hypotenuse, and $AB$ and $BC$ are the legs. 
 Step 2 (Substitute the known values): Given $AC = 5$ and $AB = 4$. Substituting these values: $5^2 = 4^2 + BC^2 \implies 25 = 16 + BC^2$ 
 Step 3 (Solve for $BC$): $BC^2 = 25 - 16 = 9 \implies BC = \sqrt{9} = 3$ 
 Step 4 (Calculate the area): The area of the right triangle is given by $\frac{1}{2} \times AB \times BC$. Substituting the known values: Area $= \frac{1}{2} \times 4 \times 3 = 6$ 
 Answer: {D} 
 Your final answer must be one of A, B, C, or D, and it should be placed within { } |

Table 7: The prompts used for choice questions in the evaluation for response generation.

| | |
|---|---|
| **Original Prompt** | |

You are an assistant for solving math problems. Your input consists of a math question and images. Give your answer directly, without any intermediate steps.

**CoT (Chain of Thought)**

You are an assistant for solving math problems. Your input consists of a math question and images. Your task is to output the solution steps and the answer. The output format should be a step-by-step approach.

**CoT with 2-shot**

Example 1:
Question: If a triangle has two sides of length 3 and 4, what is the length of the hypotenuse?
Answer:
Step 1: (Mathematical theorem used: Pythagorean theorem): The Pythagorean theorem states that in a right triangle, the square of the hypotenuse is equal to the sum of the squares of the other two sides. The formula is: $c^2 = a^2 + b^2$, where $a$ and $b$ are the legs, and $c$ is the hypotenuse.
Step 2: (Substitute the known values): Given $a = 3$ and $b = 4$. Substituting these values into the formula: $c^2 = 3^2 + 4^2 = 9 + 16 = 25$.
Step 3: (Calculate the hypotenuse): Taking the square root gives: $c = \sqrt{25} = 5$.
Answer: 5
Example 2:
Question: In the right triangle ABC, AB is perpendicular to BC. It is known that $AC = 5$ and $AB = 4$. Find the area of the right triangle.
Answer:
Step 1: (Mathematical theorem used: Pythagorean theorem): We first use the Pythagorean theorem to find the length of $BC$. The formula is: $AC^2 = AB^2 + BC^2$, where $AC$ is the hypotenuse, and $AB$ and $BC$ are the legs.
Step 2: (Substitute the known values): Given $AC = 5$ and $AB = 4$. Substituting these values: $5^2 = 4^2 + BC^2 \implies 25 = 16 + BC^2$.
Step 3: (Solve for $BC$): $BC^2 = 25 - 16 = 9 \implies BC = \sqrt{9} = 3$.
Step 4: (Calculate the area): The area of the right triangle is given by $\frac{1}{2} \times AB \times BC$. Substituting the known values: Area $= \frac{1}{2} \times 4 \times 3 = 6$.
Answer: 6
Please reason step by step. Each step is placed on a new line, using the following format: Step X (Mathematical theorem/basis used): Detailed solution steps. Answer: {}

Table 8: Prompts used for single-step free-form questions in the evaluation for response generation.

| **Original Prompt** |
| --- |
| You are an assistant for solving math problems. Your input consists of a math question and images. Each problem is a multi-step problem. Give your answer directly, without any intermediate steps. |
| **CoT (Chain of Thought)** |
| You are a math problem-solving assistant. Your input is a math problem and a picture of the problem. Each problem is a multi-step problem. Your task is to output the solution ideas and answers for each step. The output format is step-by-step. |
| **CoT with 2-shot Examples** |
| Example 1: |

Question: If a triangle has two sides of length 3 and 4, (1) what is the length of the hypotenuse? (2) what is the area of this triangle?

Answer:

(1) Step 1: (Mathematical theorem used: Pythagorean theorem): The Pythagorean theorem states that in a right triangle, the square of the hypotenuse is equal to the sum of the squares of the other two sides. The formula is: $c^2 = a^2 + b^2$, where $a$ and $b$ are the legs, and $c$ is the hypotenuse.

Step 2: (Substitute the known values): Given $a = 3$ and $b = 4$. Substituting these values into the formula: $c^2 = 3^2 + 4^2 = 9 + 16 = 25$.

Step 3: (Calculate the hypotenuse): Taking the square root gives: $c = \sqrt{25} = 5$.

So the length of the hypotenuse is 5.

(2) Step 1: The area of a right triangle is half the product of its two sides.

Step 2: So the area of this triangle is $3 \times 4/2 = 6$.

So the area of this triangle is 6.

Example 2:

Question: In the right triangle ABC, AB is perpendicular to BC. It is known that $AC = 5$ and $AB = 4$. (1) Find the area of the right triangle. (2) What is the height of the hypotenuse of this right triangle?

Answer:

(1) Step 1: (Mathematical theorem used: Pythagorean theorem): We first use the Pythagorean theorem to find the length of $BC$. The formula is: $AC^2 = AB^2 + BC^2$, where $AC$ is the hypotenuse, and $AB$ and $BC$ are the legs.

Step 2: (Substitute the known values): Given $AC = 5$ and $AB = 4$. Substituting these values: $5^2 = 4^2 + BC^2 \implies 25 = 16 + BC^2$.

Step 3: (Solve for $BC$): $BC^2 = 25 - 16 = 9 \implies BC = \sqrt{9} = 3$.

Step 4: (Calculate the area): The area of the right triangle is given by $\frac{1}{2} \times AB \times BC$. Substituting the known values: Area $= \frac{1}{2} \times 4 \times 3 = 6$.

So the area of the right triangle is 6.

(2) Step 1: According to the equal area method, the area of a right triangle is equal to half the product of the two right-angled sides, which is also equal to half the product of the hypotenuse and the corresponding height.

Step 2: According to the above principle and the conclusion of the first step, we can get $AB \times BC/2 = AC \times h/2$.

Step 3: Substituting the values, we get $h = 3 \times 4/5 = 2.4$.

So the height of the hypotenuse of this right triangle is 2.4.

Please reason step by step. Each step is placed on a new line, using the following format: Step X (Mathematical theorem/basis used): Detailed solution steps. Answer:{}

Table 9: Prompts used for multi-step free-form questions in the evaluation for response generation.

| **Multiple-Choice Prompt** |
| --- |
| You are an assistant for evaluating math problems. Your task is to extract the model's answer to the given multiple-choice question and compare it with the ground truth. |
| Steps: |
| 1. Extract the model's answer. The answer must be one of A, B, C, or D. |
| 2. Compare the extracted answer with the ground truth. |
| 3. Indicate whether the model's answer is correct or incorrect. |
| Output format: |
| - Extracted Answer: {A}, {B}, {C}, or {D}. |
| - Correctness: [true/false]. |
| **Single-Step Free-Form Prompt** |
| You are an assistant for evaluating math problems. Your task is to extract the model's answer to the given single-step free-form question and compare it with the ground truth. |
| Steps: |
| 1. Extract the model's final answer. |
| 2. Compare the extracted answer with the ground truth. |
| 3. Indicate whether the model's answer is correct or incorrect. |
| Output format: |
| - Extracted Answer: [Final Answer]. |
| - Correctness: [true/false]. |
| **Multi-Step Free-Form Prompt** |
| You are an assistant for evaluating math problems. Your task is to extract the model's answers to each sub-question of a multi-step free-form problem and compare them with the ground truth. |
| Steps: |
| 1. Extract the final answers for each sub-question. |
| 2. Compare the extracted answers with the corresponding ground truth. |
| 3. Indicate whether each answer is correct or incorrect. |
| Output format: |
| - Sub-Question 1: Extracted Answer: [Answer]. Correctness: [true/false]. |
| - Sub-Question 2: Extracted Answer: [Answer]. Correctness: [true/false]. |

Table 10: Prompts used for evaluating different types of math problems with the Deepseek API.

| Model | Source | URL |
|---|---|---|
| Deepseek-V3 | Deepseek-V3-0324 | `https://api-docs.deepseek.com/` |
| Math-LLaVA-13B | local checkpoint | `https://huggingface.co/Zhiqiang007/Math-LLaVA` |
| LLaVA-v1.5-7B | local checkpoint | `https://huggingface.co/liuhaotian/llava-v1.5-7b` |
| InternLM-XComposer2.5-VL-7B | local checkpoint | `https://huggingface.co/internlm/internlm-xcomposer2d5-7b` |
| Deepseek-VL2-7B | local checkpoint | `https://huggingface.co/deepseek-ai/deepseek-vl2` |
| LLaVA-NeXT-Interleave-7B | local checkpoint | `https://huggingface.co/lmms-lab/llava-next-interleave-qwen-7b` |
| LLaVA-OneVision-Chat-7B | local checkpoint | `https://huggingface.co/lmms-lab/llava-onevision-qwen2-7b-ov-chat` |
| Qwen2.5VL-Instruct-7B | local checkpoint | `https://huggingface.co/Qwen/Qwen2.5-VL-7B-Instruct` |
| LLaVA-OneVision-Chat-72B | local checkpoint | `https://huggingface.co/lmms-lab/llava-onevision-qwen2-72b-ov-chat` |
| InternVL3-8B | local checkpoint | `https://huggingface.co/OpenGVLab/InternVL3-8B` |
| Mistral-small-3.1-24b-instruct | local checkpoint | `https://huggingface.co/mistralai/Mistral-Small-3.1-24B-Instruct-2503` |
| InternVL3-78B | local checkpoint | `https://huggingface.co/OpenGVLab/InternVL3-78B` |
| Llama-4-Maverick-17B-128E | local checkpoint | `https://huggingface.co/meta-llama/Llama-4-Maverick-17B-128E-Instruct` |
| LlamaV-o1-11B | local checkpoint | `https://huggingface.co/omkarthawakar/LlamaV-o1` |
| LLaVA-CoT-11B | local checkpoint | `https://huggingface.co/Xkev/Llama-3.2V-11B-cot` |
| VLM-R1-3B | local checkpoint | `https://huggingface.co/omlab/VLM-R1-Qwen2.5VL-3B-Math-0305` |
| R1-Onevision-7B | local checkpoint | `https://huggingface.co/Fancy-MLLM/R1-Onevision-7B` |
| Vision-R1-7B | local checkpoint | `https://huggingface.co/Osilly/Vision-R1-7B` |
| Skywork-R1V2-38B | local checkpoint | `https://huggingface.co/Skywork/Skywork-R1V2-38B` |
| QvQ-72B-Preview | local checkpoint | `https://huggingface.co/Qwen/QVQ-72B-Preview` |
| GPT-4V | gpt-4-vision-latest | `https://platform.openai.com/` |
| Claude-3.5-Sonnet | claude-3.5-sonnet-2024-05-24 | `https://www.anthropic.com/news/claude-3-5-sonnet` |
| Gemini-1.5-Pro | gemini-1.5-Pro-latest | `https://ai.google.dev/` |
| Gemini-2.5-Pro | gemini-2.5-pro-preview-05-06 | `https://ai.google.dev/` |
| GPT-4o | gpt-4o-2024-05-14 | `https://platform.openai.com/` |
| Claude-3.7-Sonnet | claude-3.7-sonnet-latest | `https://www.anthropic.com/news/claude-3-7-sonnet` |
| OpenAI-o1 | OpenAI-o1-latest | `https://platform.openai.com/` |

Table 11: The source of the models used in the evaluation.

## Measurement of Solid Geometric Forms

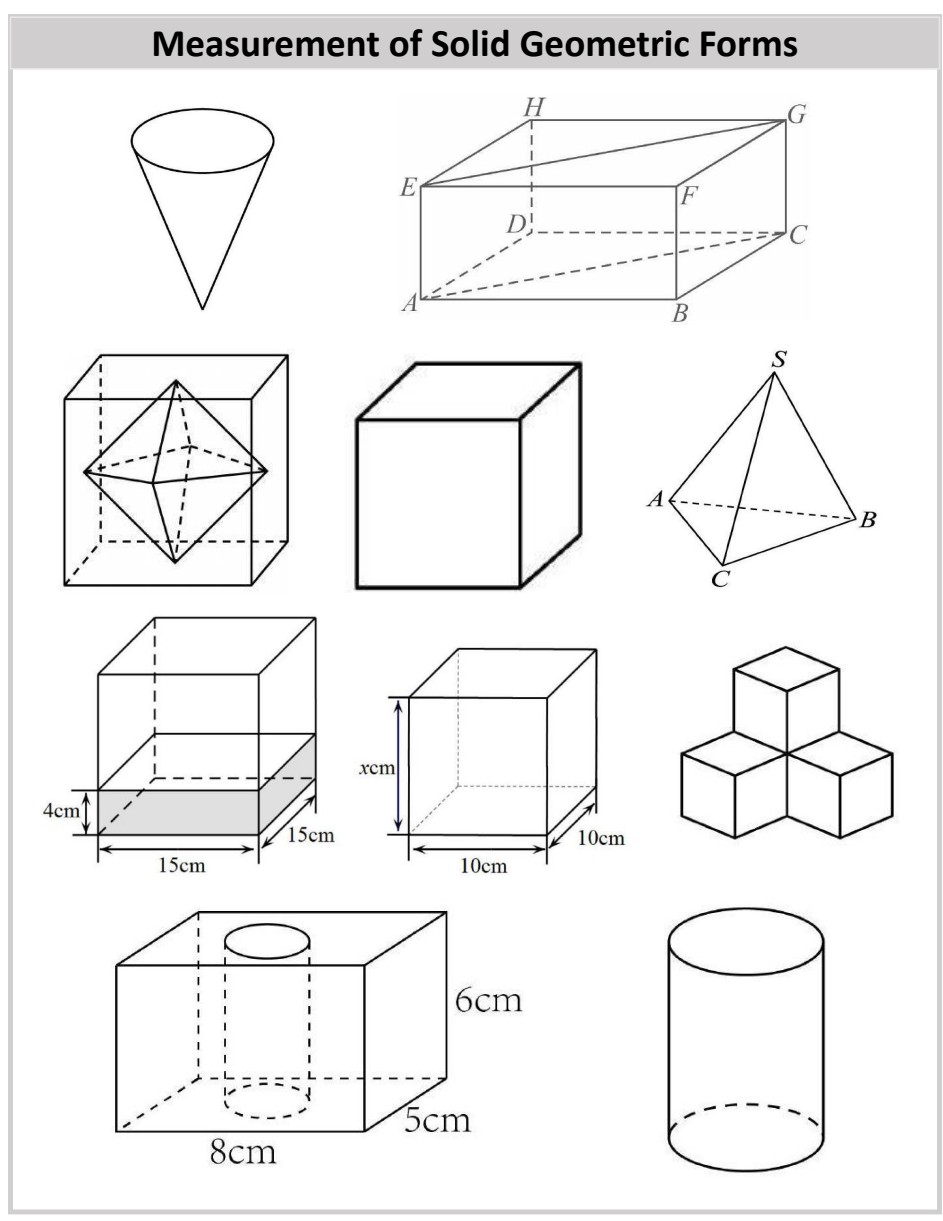

Figure 22: Some images from Measurement of Solid Geometric Forms.

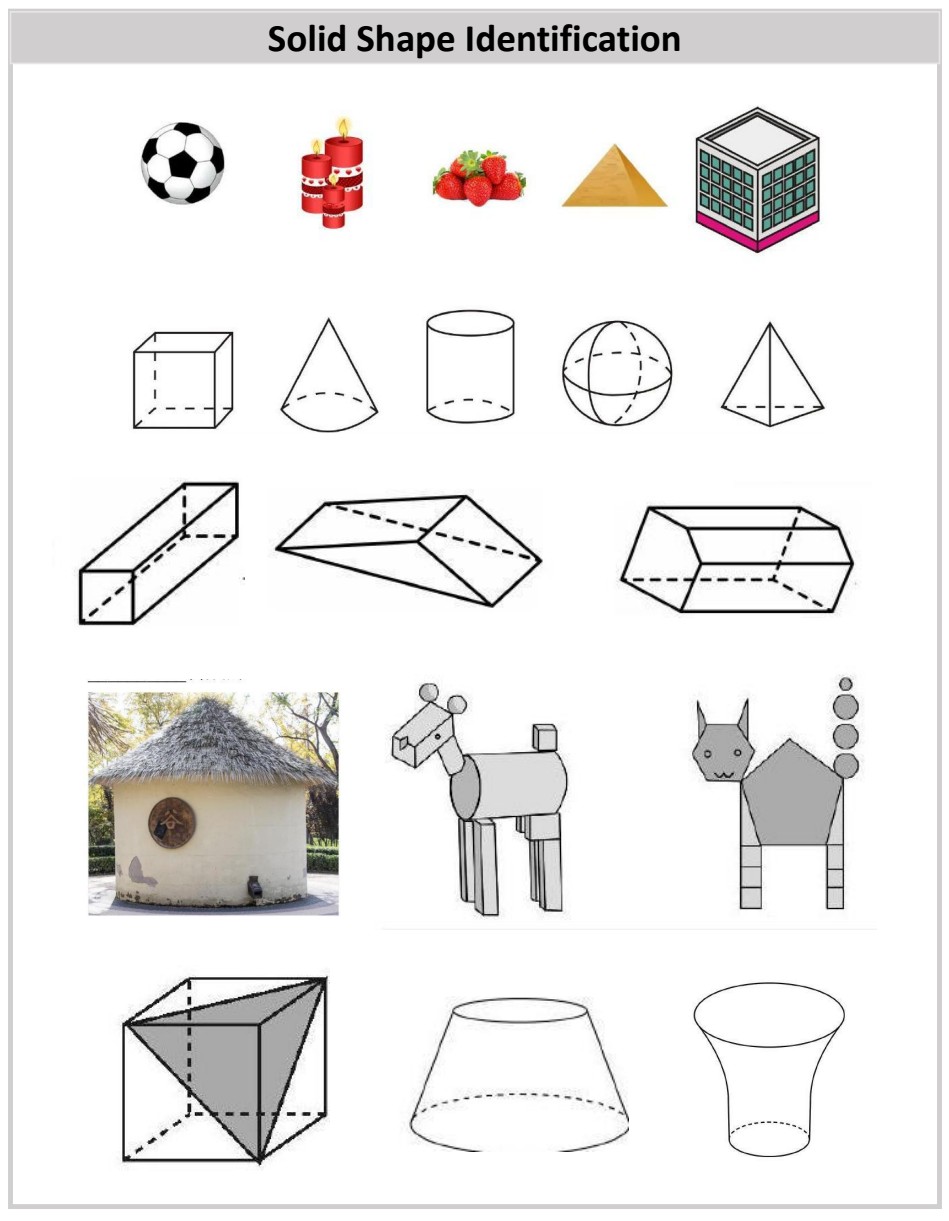

Figure 23: Some images from Solid Shape Identification.

# Spatial Metric Relations

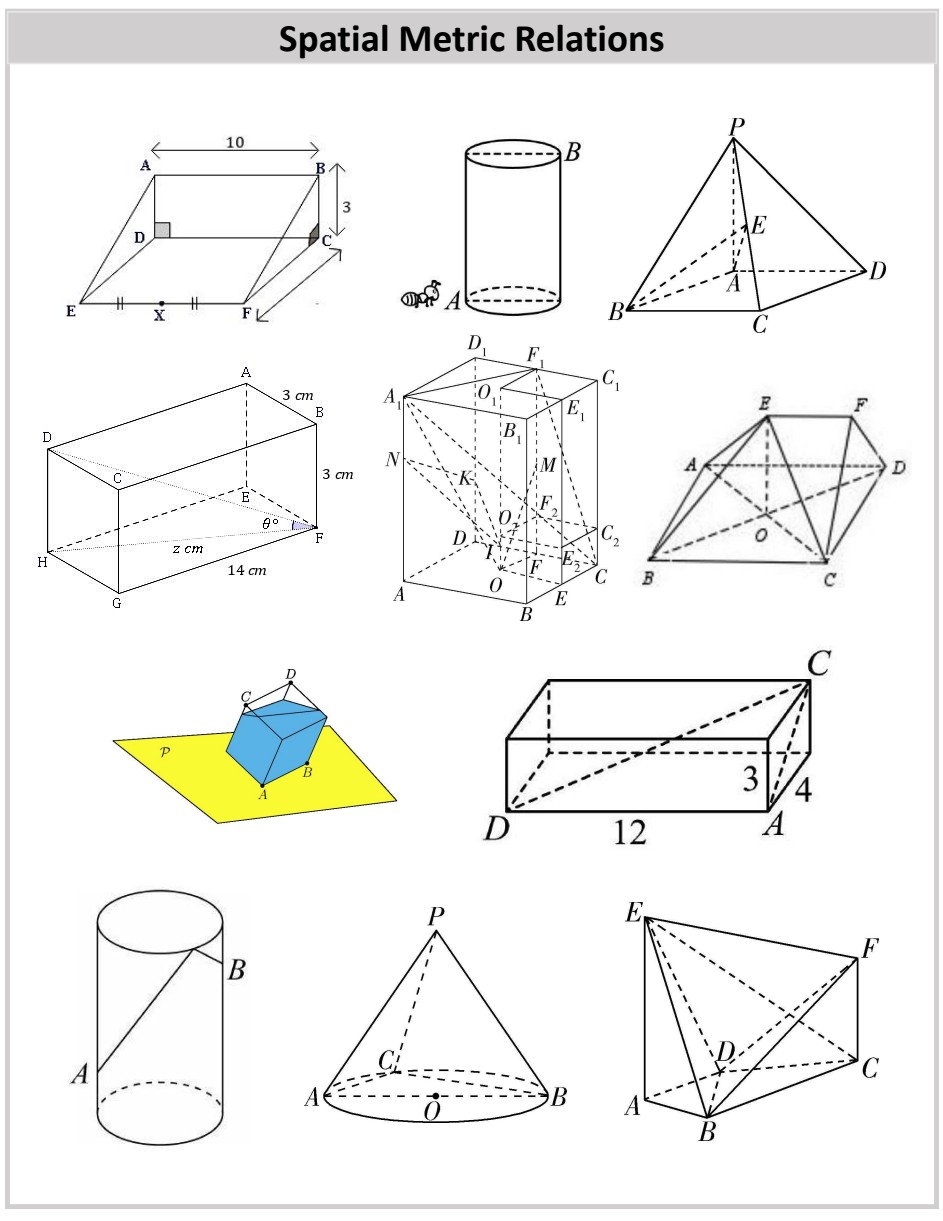

Figure 24: Some images from Spatial Metric Relations.

# Multi-view Projection

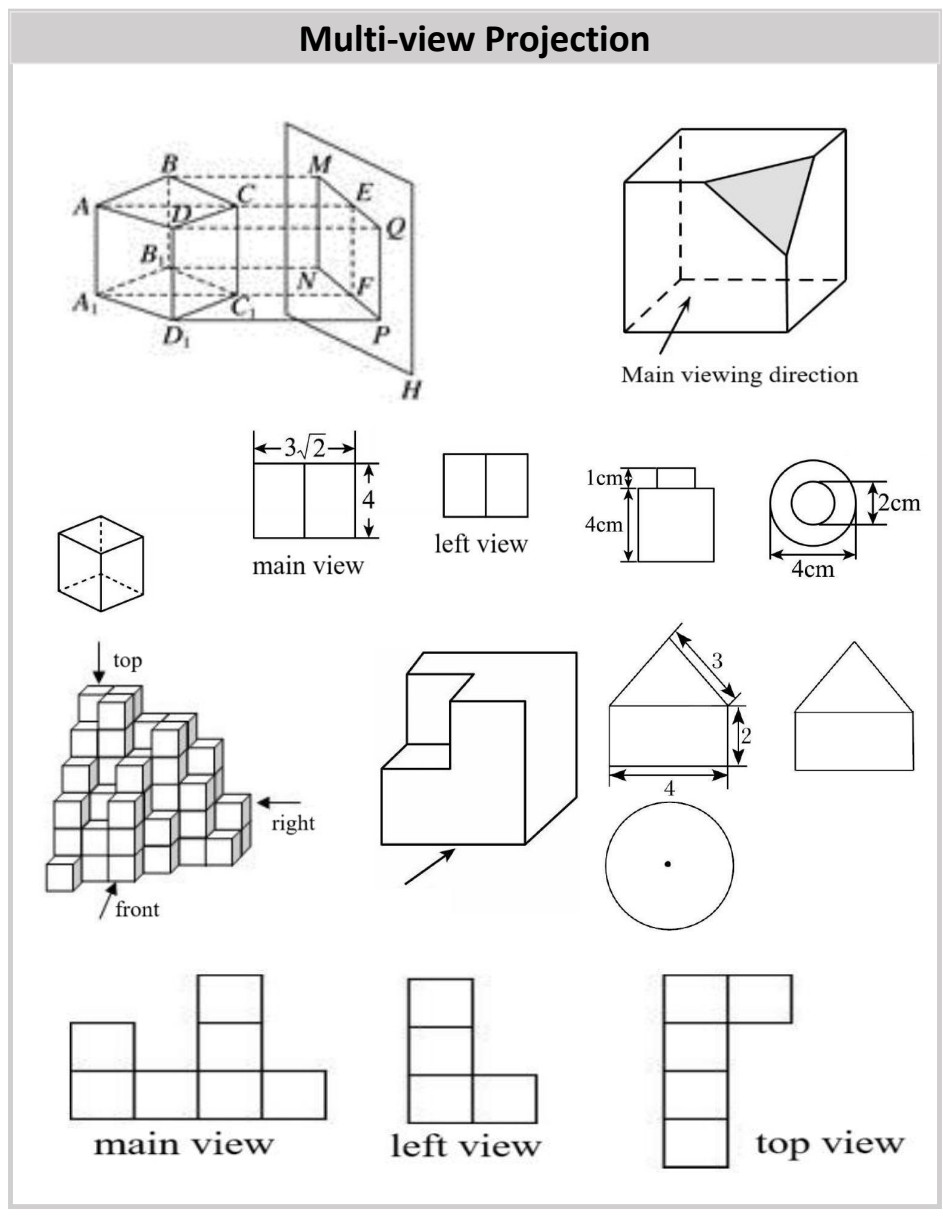

Figure 25: Some images from Measurement of Multi-view Projection.

# Planar Unfolding and Configuration

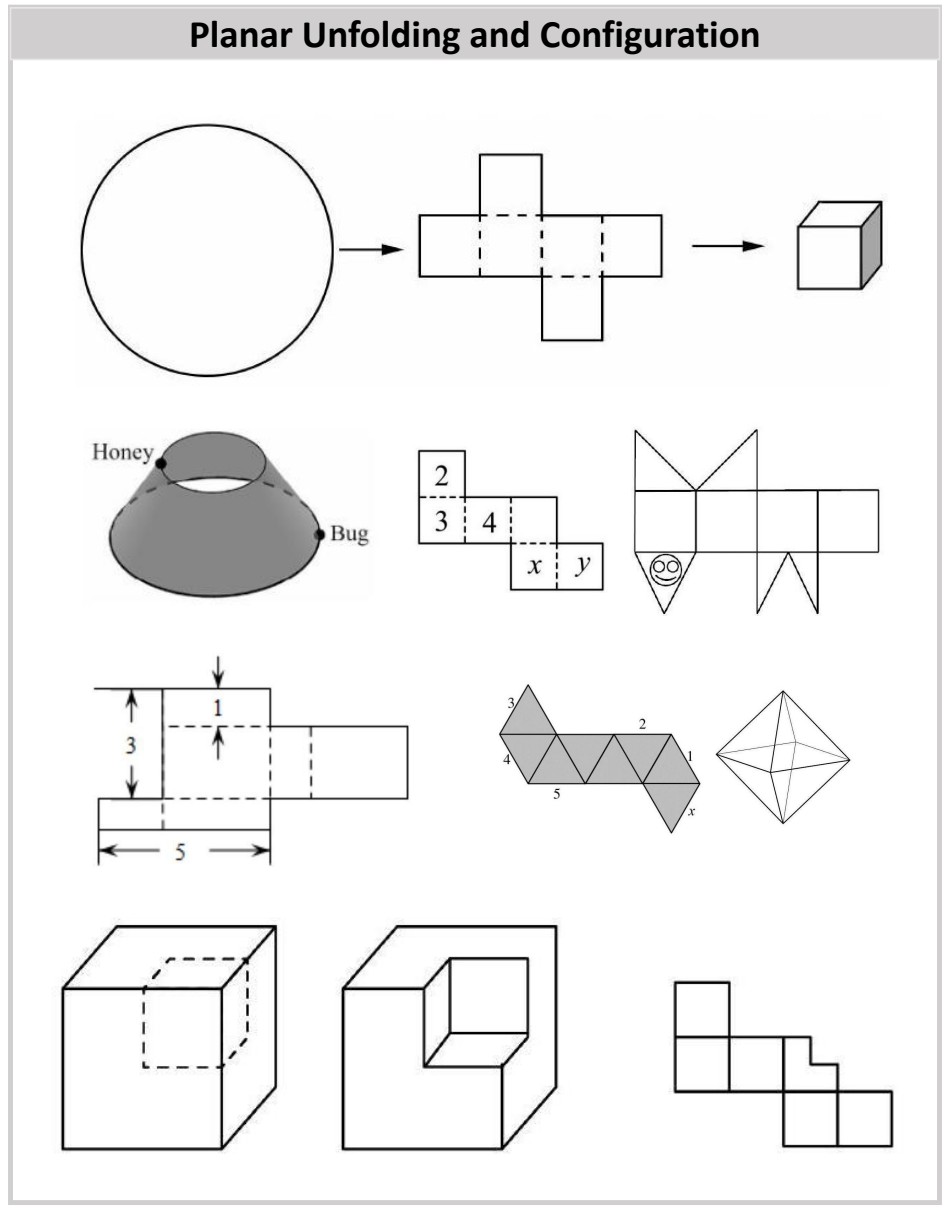

Figure 26: Some images from Planar Unfolding and Configuration.

# Composite Solid Structure

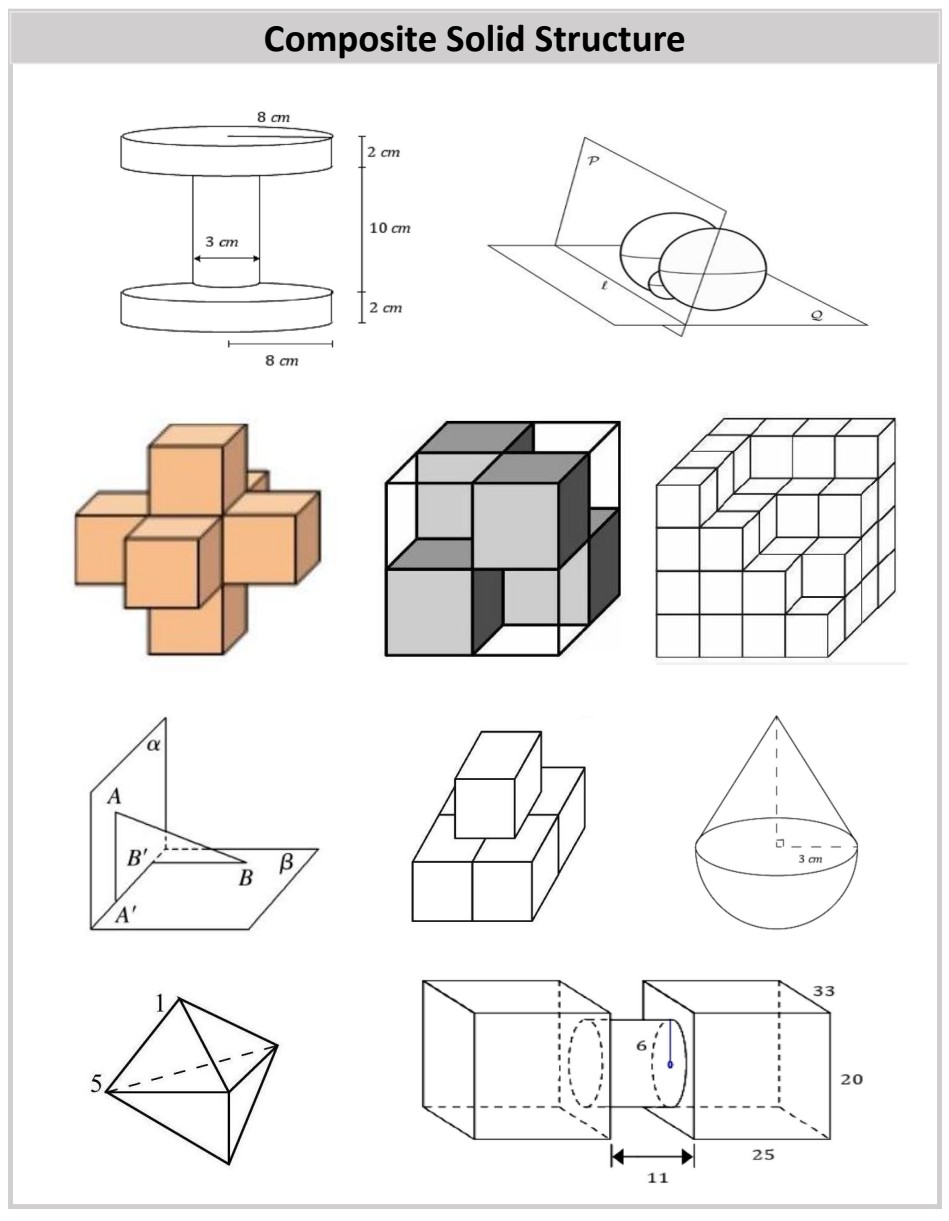

Figure 27: Some images from Composite Solid Structure.

# 3D Coordinate and Vector Reasoning

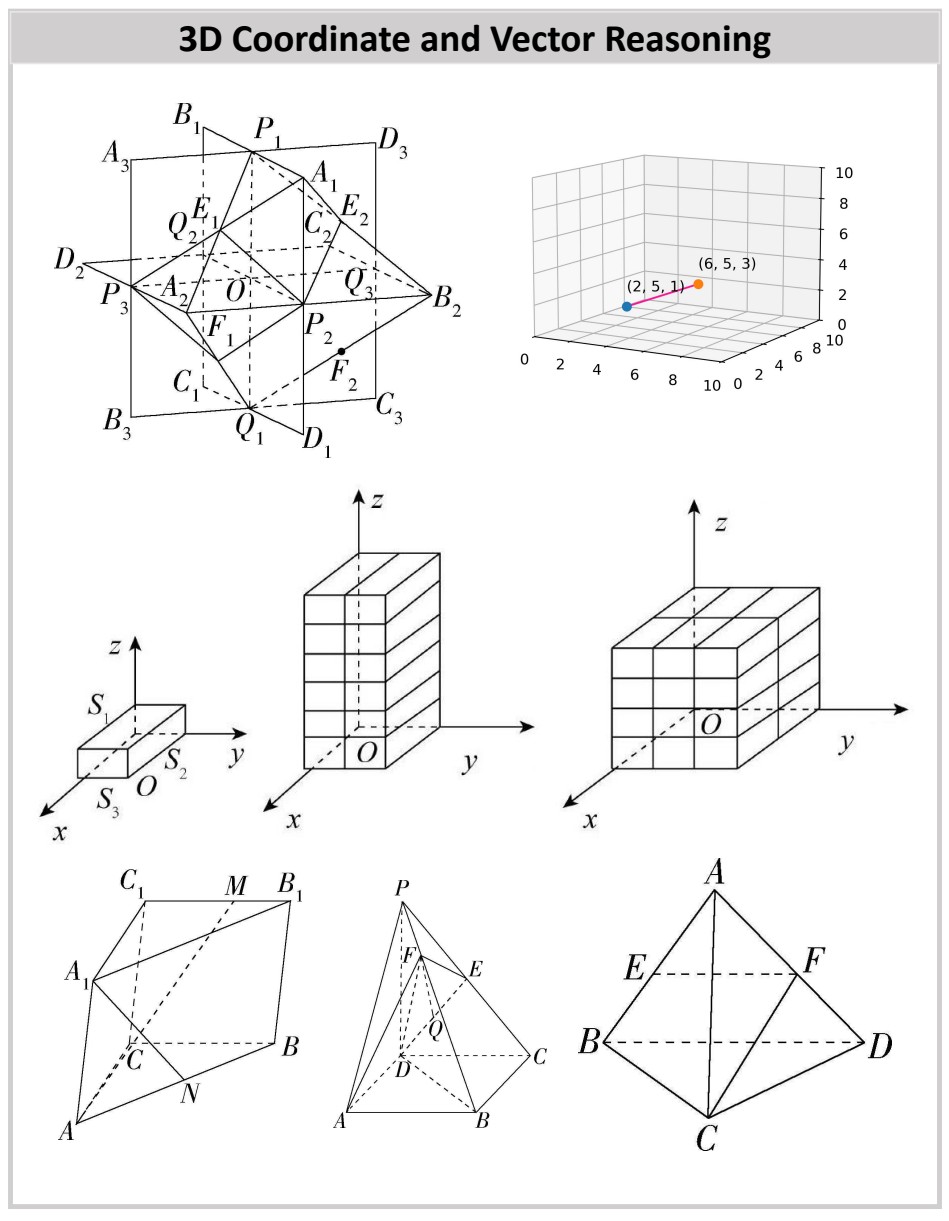

Figure 28: Some images from 3D Coordinate and Vector Reasoning.

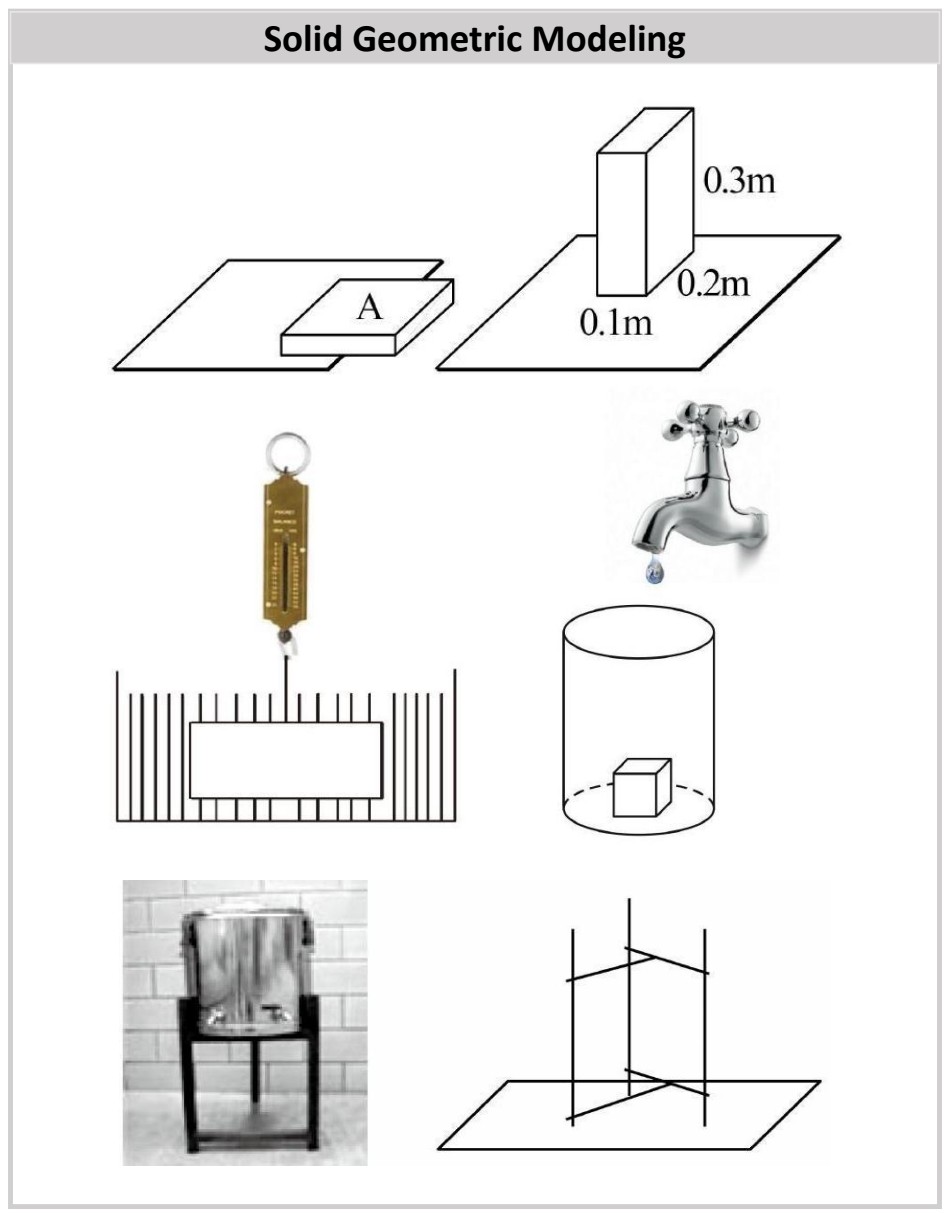

Figure 29: Some images from Measurement of Solid Geometric Modeling.

# H   Case Study

To delve into the failure cases of models, we detailed five typical error types in Table 12. Furthermore, to facilitate a better understanding of each error type, we provide examples of each error made by Gemini-2.5-pro from Figure 30 to Figure 34.

Table 12: Detailed Descriptions of Error Types.

| Error Type | Explanation |
|---|---|
| Reason Error | Errors that occur in the logical reasoning process while using knowledge concepts to solve the problem step by step. |
| Visual Perception Error | Errors in visual perception, where the model incorrectly identifies shapes or numbers in an image. |
| Knowledge Error | For a specific knowledge concept, the model is unclear or confused about it, or it misuses another knowledge concept to solve the problem. |
| Hallucination | The thought process introduces factors that are not consistent with the facts, which are not mentioned in the context of the image or question. |
| Calculation Error | Errors arising from incorrect arithmetic operations, such as multiplication, addition, or formula application, despite having the correct reasoning path. |

---

**Calculation Error**

**Question:** There is a cylindrical wooden block buried in the wall (as shown in Figure (1)). Its diameter is unknown, so it is sawed open (along the cross section) with a saw. (As shown in Figure (2) When the measured depth $CE$ is 1 inch, the sawed width $AB$ is 1 foot, and the diameter $CD$ of the wooden block is _____ inches. (1 foot = 10 inches)

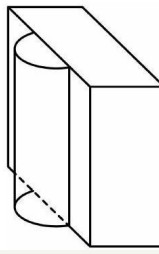 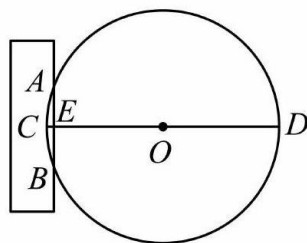

**Ground Truth: 12.16**

**Gemini-2.5-pro Answer**

Step 1: Recognize that triangle $\triangle CDE$ is a right triangle with $CE=1$ inch (depth), $AB=12$ inches (sawed width), so $AE=6$ inches, i.e., the length of $AE$ is half of $AB$.
Step 2: Use the Pythagorean theorem to compute the radius $r$ of the circle: $r^2 = CE^2 + AE^2 = 1^2 + 6^2 = 1 + 36 = 37$. So, $r = \sqrt{37}$
Step 3: The diameter is $2r = 2 \times \sqrt{37} \approx 12.12$ inches.
Answer: 12.12 inches

Figure 30: An example of calculation error.

## Reasoning Error

**Question:** An object is composed of multiple small cubes with edge lengths of 1. The shapes of the geometric body as seen from the front, left, and top are shown in the figures below. Then, the volume of the geometric body is _____.

View from the front     View from the left     View from the top

**Ground Truth: 8**

**Gemini-2.5-pro Answer**

To find the volume of the geometric body composed of small cubes with edge lengths of 1, we need to analyze the views from the front, left, and top.
Step 1 (Analyze the front view):
- The front view shows a structure with 3 columns.
- The left and right columns each have 1 cube. The middle column has 2.
  Step 2 (Analyze the left view):
- The left view shows a structure with 2 columns. The left column has 1 cube. The right column has 2 cubes.
Step 3 (Analyze the top view):
- The top view shows a structure with 5 cubes arranged in a cross pattern.
- This confirms the layout seen from the front and left views.
Step 4 (Determine the number of cubes):
- From the front view, we have 1 cube in the left column, 2 cubes in the middle column, and 1 cube in the right column.
- From the left view, we have 1 cube in the left column and 2 cubes in the right column. Combining these views, we can deduce the following:
- The left column has 1 cube. The middle column has 2 cubes.
- The right column has 1 cube. The top view confirms that there are 6 cubes in total.
Step 5 (Calculate the volume): Since each cube has an edge length of 1, the volume of each cube is $1^3 = 1$. The total volume is the number of cubes multiplied by the volume of each cube.
Answer: The volume of the geometric body is 5.

Figure 31: An example of reasoning error.

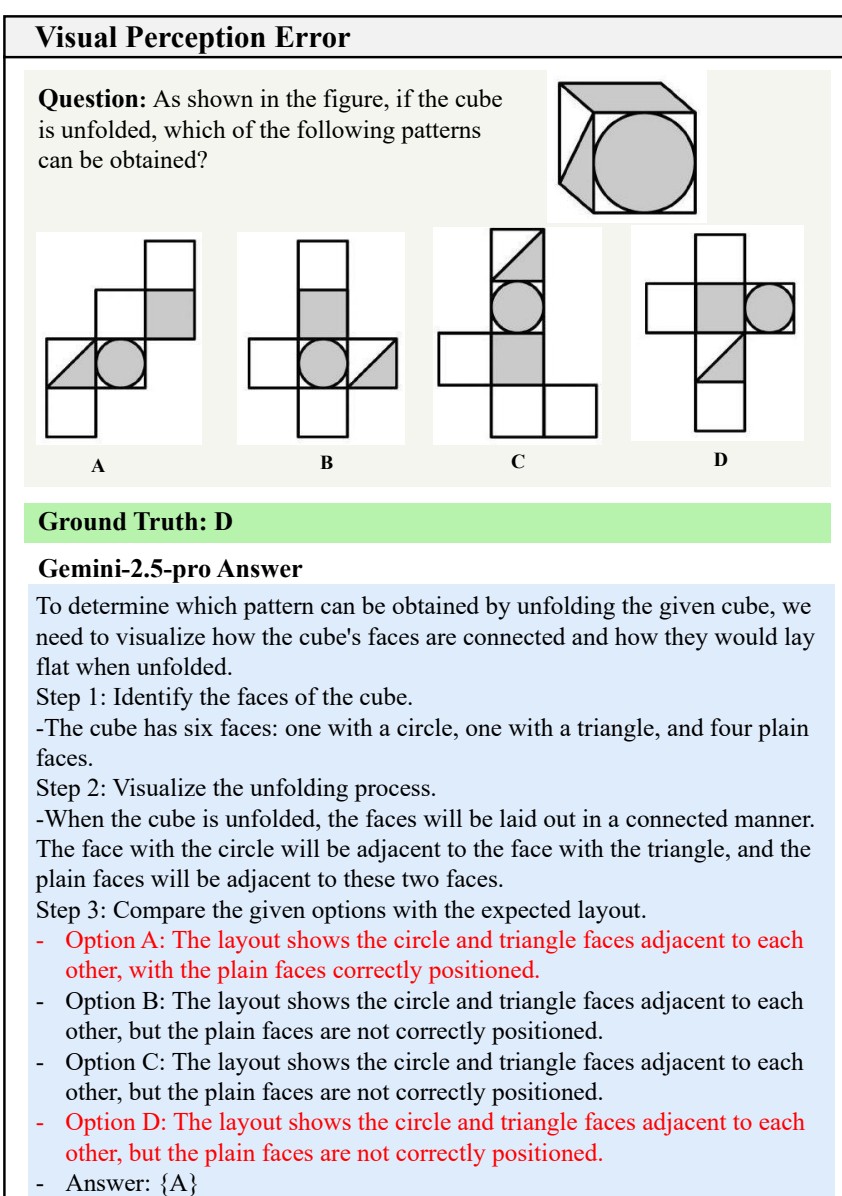

Figure 32: An example of visual perception error.

## Knowledge Error

**Question:** A regular icosahedron is a 20-faced solid where each face is an equilateral triangle and five triangles meet at every vertex. The regular icosahedron shown below has one vertex at the top, one vertex at the bottom, an upper pentagon of five vertices all adjacent to the top vertex and all in the same horizontal plane, and a lower pentagon of five vertices all adjacent to the bottom vertex and all in another horizontal plane. Find the number of paths from the top vertex to the bottom vertex such that each part of a path goes downward or horizontally along an edge of the icosahedron.

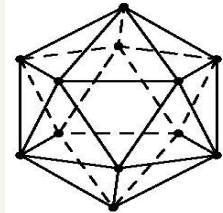

**Ground Truth: 810**

**Gemini-2.5-pro Answer: 1563**

Step 1: The problem asks for the number of paths from the top vertex to the bottom vertex in a regular icosahedron. The paths must go downward or horizontally.

Step 2: Since the icosahedron has 20 faces and 12 vertices, and each face is an equilateral triangle, we assume that the graph formed by the vertices and edges is regular with degree 5.

Step 3: Treat this as a 5-step traversal problem over a regular graph from one vertex to another. At each step, the vertex has 5 options (since degree is 5), so total possible paths $\approx 5^5 = 3125$.

Step 4: Since only downward or horizontal paths are allowed, we restrict to half of these, so approximate total $= \frac{3125}{2} = 1562.5 \approx 1563$.

Figure 33: An example of knowledge error.

## Hallucination Error

**Question**: As shown in Figure 1, this is a Rubik's Cube composed of 8 equally sized smaller cubes, with a total volume of 64.

(1)Determine the edge length of the Rubik's Cube.

(2)In the figure, quadrilateral ABCD is a square. Find the area and the side length of this square.

(3)Place the square ABCD on the number line as shown in Figure 2, such that point A coincides with -1, point E coincides with 1, and point F is symmetric to point D with respect to point E. Then, the number represented by point D on the number line is _.

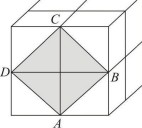 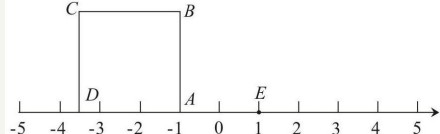

**Ground Truth:** (1) 4 (2 )8 2√2 (3) -1-2√2

**Gemini-2.5-pro Answer**

(1) The total volume of the Rubik's Cube is 64 and it consists of 8 equal smaller cubes. So, each small cube has a volume of $\frac{64}{8} = 8$, hus its edge length is $\sqrt[2]{8} = 2$. The entire cube is 2×2×2, so the cube's edge length is 2×2=4.

(2) The square ABCD lies on the face diagonal of the cube. The diagonal AC is $\sqrt{4^2 + 4^2} = \sqrt{32} = 4\sqrt{2}$. Since ABCD is a square with diagonal $4\sqrt{2}$, its side length is $\frac{4\sqrt{2}}{\sqrt{2}} = 4$, and area is $4^2 = 16$.

(3) To determine point D on the number line, we consider rotational symmetry as illustrated in a cube net. Since ABCD is placed on the number line and the cube has 8 small cubes, we assume it folds along a 3D diagonal. In 3D rotation, D corresponds to a mirrored projection over the cube center, and since E is 1, the center is at 0. Thus, point D =0 - (3-0) = -3.

So the answer is -3.

Figure 34: An example of hallucination error.

