# OpenReview forum: "SolidGeo: Measuring Multimodal Spatial Math Reasoning in Solid Geometry"
_NeurIPS.cc/2025/Datasets_and_Benchmarks_Track — NeurIPS 2025 Datasets and Benchmarks Track poster_

### Official Review · Reviewer_2H7L · 2025-06-05

**Rating:** 5
**Confidence:** 3

**Summary:**

The submission introduces SOLIDGEO, the first large-scale benchmark specifically designed to evaluate multimodal large language models (MLLMs) on solid geometry mathematical reasoning. The dataset contains 3,113 problems derived from both real-world K–12 and competition-level sources, with visual contexts, difficulty annotations, and fine-grained subcategories across eight solid geometry topics. Extensive experiments involving 27 models are conducted.

**Additional Feedback:**

None.

**Dataset Code Accessibility:**

Yes

**Dataset Code Comments:**

The dataset is openly available at HuggingFace and appears well-documented. The dataset includes images, answers, and annotations for category and difficulty, making it complete and ready to use.

**Ethical Comments:**

None.

**Ethical Considerations:**

No, there are no or only very minor ethics concerns

**Final Justification:**

My questions were all well answered by the author, so I’m keeping my rating.

**Limitations Weaknesses:**

1. The absence of a formalized symbolic language for representing solid geometry problems, hindering standardized model evaluation. The authors are encouraged to have a deeper analysis of this problem or provide a solution to this problem in future work.
2. The error analysis part is somewhat too simple. It is encouraged to have a more detailed error analysis that can make the improvement of the model more directional.
3. The dataset contains only English and Chinese. Is it possible to include more languages in the future?

**Strengths Contributions:**

1. The proposed dataset fills a gap in the evaluation of MLLMs by focusing on solid geometry.

2. With over 3,000 multimodal problems, the proposed dataset is diverse in both format (multiple-choice, single-/multi-step) and content (projection, 3D vectors, unfolding, etc.).

3. The paper conducts an extensive empirical study over 27 MLLMs using zero-shot, CoT, and few-shot prompts, identifying trends in performance, token usage, and failure modes

4. Error analysis reveals that reasoning and visual perception errors dominate, informing future directions in model training and architecture.

---

> ### Author Rebuttal · Authors · 2025-07-31
>
> Dear Reviewer 2H7L:
>
> Thank you for your kind words and positive feedback! Your approval is the great encouragement for us and motivates us to continue advancing our work.  Below, we meticulously provide responses to each of your comments and outline the modifications made to the manuscript.
>
> > **Q1: The absence of a formalized symbolic language for representing solid geometry problems**
>
> Thank you for pointing out the lack of a formalized symbolic representation for solid geometry problems. We fully agree that such a language would significantly improve the standardization and interpretability of model evaluation. In fact, this limitation is acknowledged in the *Limitations and Future Work* section of our paper.
>
> We would like to emphasize that developing such a symbolic language is a direction we are already actively exploring. The construction of the SOLIDGEO benchmark serves as a foundational step toward this goal. By systematically organizing solid geometry problems, our benchmark paves the way for future work on designing a formalized representation that captures geometric entities, spatial relationships, and reasoning steps more precisely.
>
> We believe this benchmark can stimulate further research on formalizing spatial reasoning tasks, and we are committed to contributing to that effort in our future extensions.
>
>
>
> > **Q2: The error analysis part is somewhat too simple**
>
> Thank you for your helpful comment. Our error analysis includes five distinct error types, namely: *Visual Perception Error*, *Reasoning Error*, *Knowledge Error*, *Calculation Error*, and *Hallucination*. Due to space limitations, we only presented one representative example (a reasoning error) in the main text. However, a broader and more detailed error taxonomy, along with additional examples, is provided in **Appendix H**.
>
> To improve clarity, we will consider summarizing these error types more explicitly in the main paper. We will also include more illustrative error case analyses in the final version, which we believe will offer concrete and actionable insights for future work on model architecture design and training strategies.
>
>
>
> > **Q3: The dataset contains only English and Chinese. Is it possible to include more languages in the future?**
>
> Thank you for your thoughtful suggestion. We fully agree that multilingual support would enhance the utility of SOLIDGEO for a broader range of users and research contexts.
>
> That said, collecting high-quality multimodal problems in solid geometry remains a significant challenge. Even after extensive efforts to combine data from both existing sources and newly collected problems—followed by de-duplication and strict quality filtering—SOLIDGEO currently contains only **3,113 high-quality examples**. This reflects the scarcity and construction cost of such specialized data.
>
> Looking ahead, we see two possible paths for multilingual expansion:
>
> 1. **Sourcing native-language solid geometry problems from multilingual educational contexts**. While ideal, this approach would require access to diverse curricula and annotated visual materials, which is unfortunately beyond our current resource scope.
> 2. **Translating the current English and Chinese content into other languages**. This is a more feasible and scalable direction for us, and we are actively considering such efforts for future versions of SOLIDGEO.
>
> Given the above, we believe the second approach—multilingual translation of our existing dataset—is the more realistic and practical path forward in the near term. We appreciate your suggestion and will prioritize this direction as part of our roadmap for dataset expansion.

---

### Official Review · Reviewer_WsUt · 2025-06-16

**Rating:** 5
**Confidence:** 5

**Summary:**

The paper introduces SOLIDGEO, a new benchmark for evaluating multimodal LLMs (MLLMs) on 3D solid geometry problems—an area that’s been underexplored compared to 2D geometry even though it requires deeper spatial understanding. It addresses limitations of existing datasets (like MathVista, MathVision, MathVerse), which include few solid-geometry problems and mostly basic tasks such as counting or volume estimation. Extensive experiments are conducted for various MLLMs, including both open-source (e.g., LLaVA, InternVL, Qwen-VL) and closed-source systems (GPT-4V, Claude‑3.7, Gemini‑2.5, OpenAI‑o1) .

**Dataset Code Accessibility:**

Yes

**Dataset Code Comments:**

The authors have provided huggingface link to dataset and github link to code.

**Ethical Considerations:**

No, there are no or only very minor ethics concerns

**Final Justification:**

I appreciate the author's detailed responses, which addresses most of my concerns. I strongly recommend that the authors include the additional details into their final version. I think this work presents a valuable asset for solid geometry, which will be useful for the community. I have updated my score accordingly.

**Limitations Weaknesses:**

- Human performance is reported (77.5%), but details are sparse: we don’t get a full breakdown of human error types, or inter-annotator agreement.
- Regarding the labelling of difficulty level, the explanation in line 177 - 186 seems quite vague. How exactly did the authors perform voting? How many times are the models inferences, and how did the authors determine the difficulty levels? I may have missed the detail, please correct me if there are other places mentioning this detail.
- For the analysis in Difficulty Levels between line 266 to line 277, the authors found that most open source models show a decline in accuracy as the level of difficulty increases, and this phenomenon does not apply for thinking models like Gemini-2.5-pro and OpenAI-o1, would it be because the difficulty levels are labelled using the other models, but not these two, would this indicate that the labeling of difficulty contains bias?
- While SOLIDGEO focuses on text + 2D images, it does not incorporate 3D visual inputs like meshes, point clouds, or interactive diagrams, which are more natural representations for solid geometry. This limits the benchmark’s ability to fully capture the spatial reasoning potential of MLLMs beyond static 2D projections.

**Strengths Contributions:**

- Comprehensive benchmark: First large-scale, fine‑categorized dataset focused solely on solid geometry.

- New evaluation perspective: Offers deeper insights into MLLM spatial reasoning limits, contrasting model types and prompting styles.

- Guiding future research: Reveals core challenges—especially visual grounding, spatial understanding, and efficient reasoning in 3D contexts.

- Future directions: Authors highlight the need for more language diversity, formal symbolic representation for 3D, and new strategies to reduce hallucinations

---

> ### Author Rebuttal · Authors · 2025-07-31
>
> Dear Reviewer WsUt:
>
> Thank you for your detailed and constructive review. We sincerely appreciate the time and effort you devoted to evaluating our work. Below, we provide point-by-point responses to your comments and clarify the corresponding aspects of our study.
>
> > **Q1: More details about Human Performance**
>
> Thank you for raising this important point. To better understand human performance beyond the overall accuracy of 77.5%, we conducted a detailed analysis of human errors.
>
> We randomly sampled **300 incorrect responses** from the human evaluation and categorized them into three main error types:
>
> - **Reasoning Errors (56.5%):** This was the most common error type, where students failed in the logical deduction process. For example, they incorrectly inferred spatial relationships from a 2D projection or made a mistake in a multi-step reasoning. This highlights that the core challenge of SOLIDGEO, even for humans, lies in complex spatial reasoning.
> - **Knowledge Error (37.0%):** These errors occurred when students misunderstood, forgot, or misapplied a specific geometric theorem or formula.
> - **Calculation Errors (6.5%):** This was the least frequent error type, involving simple arithmetic mistakes or incorrect substitutions into a correct formula.
>
> This distribution highlights that the majority of mistakes stem from high-level reasoning failures, rather than simple miscalculations, reflecting the inherent difficulty of 3D geometry tasks even for trained students.
>
> In terms of inter-annotator agreement, all answers were graded by two experienced mathematics instructors. To assess the consistency of the scoring, we randomly sampled 300 answers and had them independently reviewed by two additional annotators, resulting in a 99.3% agreement rate. These details will be incorporated into the final version to enhance the credibility and transparency of the manual benchmark.
>
>
>
> > **Q2: Difficulty Level Labeling**
>
> Thank you for your question regarding the difficulty labeling process. We apologize if the details were not easy to find in the main text.
>
> The full, step-by-step description of our labeling methodology is located in the **Appendix D.3 (Data Annotation Protocol)** of our submitted manuscript. For your convenience, we summarize the key steps here:
>
> Our process was a **two-stage, model-assisted protocol**:
>
> 1. **Model-Based Voting:** We prompted three advanced MLLMs (GPT-4o, Claude-3.7-Sonnet, and Qwen-VL-Max) **with a single inference each** for every problem, using the detailed prompt provided in **Appendix Table 6**. An initial label was assigned based on a majority vote.
> 2. **Human Adjudication & Verification:** Cases with no majority were resolved by human expert. Critically, all 3,113 final labels were then reviewed and verified by our team to ensure accuracy and consistency.
>
> We hope this summary, along with the detailed descriptions already present in Appendices D.3 and Table 6, fully clarifies our rigorous labeling procedure.
>
>
>
>
>
> > **Q3: Difficulty Labeling and Potential Bias Toward Certain Models**
>
> Thank you for raising this important question. We would like to clarify that **the labeling of difficulty levels is not biased toward specific models**, and we offer two key pieces of evidence to support this claim:
>
> 1. **Human Verification:** While the initial difficulty labels were obtained via model voting, all 3,113 labels were subsequently reviewed and verified by our annotators. This ensured consistency, accuracy, and model-agnostic judgment across the dataset.
> 2. **Human Performance Correlation:** To further validate the quality and neutrality of our difficulty labels, we analyzed human accuracy across difficulty levels. We found a clear negative correlation between problem difficulty and human performance, Easy: 90.1%, Medium: 73.9%, Hard: 51.3%.
>
> This strong alignment with human perception reinforces the reliability of our annotations, independent of any specific model behavior.
>
> As for why models like **Gemini-2.5-Pro** and **OpenAI-o1** do not show declining performance with difficulty level, we believe this reflects their **overthinking tendency**, as mentioned in our paper (lines 272–273). On easier problems, these models may generate overly elaborate reasoning chains that ultimately hurt performance.
>
> We illustrate this with a concrete example from SOLIDGEO, which was labeled as an easy problem based on model voting and human verification. Despite its simplicity, Gemini-2.5-Pro answered incorrectly, while InternVL3 produced the correct answer:
>
> **Problem (ID: 903)**:
> “As shown in the figure, the height of the solid is 8 cm. Find the volume of the solid.”
> **Ground Truth**: 32 cm³
>
> - **Gemini-2.5-Pro** hallucinates a complex structure involving multiple cylinders and cones, and gets trapped in a **convoluted chain of symbolic deduction** that leads to a logical contradiction: it incorrectly derives **r² = 2r²**, concludes **r = 0**, and ultimately outputs a volume of **0 cm³**. Notably, this response spans over **1,200 tokens**, reflecting a tendency toward unnecessarily elaborate reasoning even on simple problems.
> - In contrast, **InternVL3** follows a correct and straightforward geometric approach: it identifies a triangular base of area 4 cm² and multiplies it by the height (8 cm) to accurately compute the volume as **32 cm³**, requiring only **385 tokens** in total.
>
> This case clearly demonstrates that the unexpected trend in performance is not due to difficulty labeling bias, but rather reflects a model-specific overthinking behavior, particularly for advanced models on conceptually simple problems.
>
>
>
> > **Q4: Limitation of 2D Projections — Lack of 3D Visual Inputs**
>
> Thank you for highlighting this important limitation. We fully agree that 3D visual modalities such as meshes, point clouds, or interactive diagrams represent a promising direction for better evaluating the spatial reasoning capabilities of multimodal LLMs.
>
> However, the current design of SOLIDGEO intentionally focuses on solid geometry problems as they naturally appear in **real-world educational contexts**, which are almost universally presented through **2D images and textual descriptions**. This choice aligns with the benchmark’s goal of evaluating model performance in settings that mirror how students typically encounter these problems.
>
> That said, we acknowledge that incorporating richer 3D visual modalities would enable more comprehensive assessments of spatial understanding. Due to resource limitations and the lack of publicly available 3D-annotated educational datasets, we were not able to include these modalities in the current version.
>
> Expanding SOLIDGEO to include synthetic or captured 3D representations is an exciting direction in future work, and we thank the reviewer for emphasizing its importance.

---

### Official Review · Reviewer_A3mF · 2025-06-18

**Rating:** 5
**Confidence:** 4

**Summary:**

Geometry is a fundamental branch of mathematics and plays an important role in our real world. However, the authors argue that current geometry benchmarks mainly focus on plane geometry and largely ignore solid geometry. To fill this gap, the authors introduce SolidGeo, the first large-scale benchmark to evaluate the solid geometry reasoning capabilities of MLLMs. In particular, the proposed benchmark consists of 3,113 problems, covering several sub-categories. Besides, the authors also conduct extensive experiments to analyze the 3D understanding ability of existing MLLMs in the proposed testbed.

**Additional Feedback:**

No additional feedback, please refer to the main comments.

**Dataset Code Accessibility:**

Yes

**Dataset Code Comments:**

The authors provide valid dataset/code download links in the haggingface platform.

**Ethical Comments:**

No ethical comments.

**Ethical Considerations:**

No, there are no or only very minor ethics concerns

**Final Justification:**

Overall, I am satisfied with the quality of this paper and rate this paper as '5-accept' in the first-round review. In the rebuttal phase, the authors provides more experiments and discussions. After reading the comments from other reviewers and the feedbacks from the authors, I decide to maintain my initial rating (5-accept).

**Limitations Weaknesses:**

From my perspective, this paper has no major flaws.

Minor:
- In Table 4 and Line 247-265, the authors report that CoT does not bring significant performance gains in solid geometry tasks. Providing more in-depth experiments and analysis would be better.
- Stats of K12/University/Competition problems is required.
- Maintain consistent style.  e.g. Line 84, Difficulty level -> Difficulty Level

**Strengths Contributions:**

- Good motivation. The authors carefully review existing datasets and find solid geometry problems are insufficiently covered in current benchmarks, and proposed a benchmark to fill this gap.
- This paper is well-presented.
- The author provides rich details about the dataset construction and a substantial amount of supplementary data
- The proposed benchmark is well designed, and provides an effective testbed for future work.
- Extensive experiments and insightful discussions.

---

> ### Author Rebuttal · Authors · 2025-07-31
>
> Dear Reviewer A3mF：
>
> Thank you for your positive feedback and valuable suggestions! We sincerely appreciate the time and effort you have dedicated to reviewing our work. Below, we meticulously provide responses to each of your comments and outline the modifications based on your suggestions.
>
> > **Q1: CoT does not bring significant performance gains in solid geometry tasks**
>
> Thank you for this valuable suggestion. We would like to clarify that our current version already includes a dedicated analysis of Chain-of-Thought (CoT) prompting in **Table 4**, **Figure 4**, and **Lines 247–265**. To further improve clarity, we now provide the following table summarizing the **average output length (tokens)** and **accuracy changes** under Original and CoT prompting across 10 models.
>
> | Model         | Orig. Tokens | CoT Tokens | Δ Tokens (%) | Orig. Acc (%) | CoT Acc (%) | Δ Accuracy |
> | ------------- | ------------ | ---------- | ------------ | ------------- | ----------- | ---------- |
> | Claude3.5     | 992          | 1008       | +1.6%        | 22.2          | 21.7        | –0.5       |
> | Gemini-1.5    | 1003         | 1157       | **+15.4%** ↑ | 25.3          | 26.1        | **+0.8** ✔️ |
> | GPT-4V        | 1433         | 1508       | +5.2%        | 24.2          | 23.0        | –1.2       |
> | GPT-4o        | 1344         | 1391       | +3.5%        | 25.5          | 24.9        | –0.6       |
> | LLaVA-OV-7B   | 353          | 440        | **+24.6%** ↑ | 8.6           | 10.5        | **+1.9** ✔️ |
> | Qwen2.5-7B    | 490          | 550        | **+12.2%** ↑ | 15.5          | 16.0        | **+0.5✔️**  |
> | LLaVA-OV-72B  | 396          | 396        | ±0.0%        | 15.9          | 14.8        | –1.1       |
> | InternVL3-8B  | 488          | 547        | **+12.1%** ↑ | 17.7          | 18.1        | **+0.4**✔️  |
> | Qwen2.5-72B   | 484          | 590        | **+21.9%** ↑ | 24.2          | 28.8        | **+4.6** ✔️ |
> | InternVL3-78B | 493          | 514        | +4.3%        | 26.2          | 25.8        | –0.4       |
>
> As the table shows, an increase in output tokens caused by CoT prompting correlates with improved performance only when the increase is substantial (typically exceeding ~10%). For instance, models such as **Qwen2.5-72B (+21.9%)** and **LLaVA-OV-7B (+24.6%)** demonstrate both significant token growth and accuracy improvements. In contrast, models like **GPT-4o (+3.5%)**, **Claude3.5 (+1.6%)**, and **InternVL3-78B (+4.3%)** show only modest increases in output length, with no corresponding gains—or even slight drops in accuracy.
>
> These findings suggest that a minimal increase in reasoning length may be insufficient to activate the potential benefits of CoT prompting. For spatial reasoning tasks like those in solid geometry, CoT appears to be beneficial only when it elicits deeper and more structured reasoning traces. Otherwise, it may simply introduce superficial verbosity without improving the model’s actual reasoning quality. In fact, the relevant phenomenon has been observed in MathVision [1], but it does not give a corresponding explanation.
>
> We appreciate your comment and will incorporate this extended analysis into the final version to make our findings clearer and more interpretable.
>
>
>
> > **Q2: Stats of K12/University/Competition problems is required.**
>
> Thank you for the helpful suggestion. The problems in SOLIDGEO are drawn from a diverse range of educational sources. Specifically, approximately **71.6%** of the problems originate from K–12 curricula, **24.9%** are derived from competitions, and the remaining **3.5%** come from university-level.
>
> We will include this breakdown in the final version to improve clarity.
>
>
>
> > **Q3: Maintain consistent style. e.g., Line 84, "Difficulty level" → "Difficulty Level"**
>
> Thank you for the careful observation. We will standardize the capitalization of key terms such as **"Difficulty Level"** in the final version to ensure consistent style throughout the paper. We have also reviewed other terms to maintain uniform usage.
>
> **Reference**
>
> [1]. Ke Wang, Junting Pan, Weikang Shi, Zimu Lu, Houxing Ren, Aojun Zhou, Mingjie Zhan, and Hongsheng Li. Measuring multimodal mathematical reasoning with math-vision dataset. Advances in Neural Information Processing Systems, 37:95095–95169, 2024.

---

> > ### Comment · Reviewer_A3mF · 2025-08-05
> >
> > Thank you for your rebuttal. I appreciate your experiments/discussions and have decided to maintain my original score - accept.

---

> > > ### Author Response · Authors · 2025-08-05
> > >
> > > Thank you very much for your kind response and for maintaining your positive assessment. We truly appreciate your thoughtful feedback throughout the review process, as well as your recognition of our work and rebuttal. Your insights have been valuable in helping us strengthen the final version of our paper.

---

### Official Review · Reviewer_Ab3A · 2025-06-23

**Ethics Flags:** Data privacy, copyright, and consent
**Rating:** 4
**Confidence:** 4

**Summary:**

This paper introduces SOLIDGEO, a new large-scale benchmark designed to evaluate the solid geometry reasoning capabilities of the Multimodal Large Models. The authors identify a gap in existing math benchmarks for MLLMs, which mostly focus on plane geometry rather than the more complex spatial reasoning required for 3D problems. SOLIDGEO consists of 3000-plus real-world problems from K-12, University, and competition sources, each annotated with visual context, difficulty level, and one of eight fine-grained solid geometry classes.

The authors conduct an extensive evaluation of 27 MLLMs, revealing that even state-of-the-art models struggle with solid geometry, performing far below than the human baseline. The detailed analysis of model performance across different categories, difficulty levels, error types provides value insights into the current limit of MLLMs in spatial intelligence.

**Additional Feedback:**

Based on the strengths and limitations of the work, I recommend a borderline accept for this paper. It addresses an important and timely need in the evaluation of Multimodal Large Language Models (MLLMs).

The work represents a promising first step. However, to enhance its impact and robustness, I suggest incorporating a broader range of real-world geometry problems, and creating new, original problems that are not publicly available. This would help ensure that the benchmark remains challenging and less likely to overlap with data that MLLMs may have already seen during pretraining.

**Dataset Code Accessibility:**

Yes

**Dataset Code Comments:**

The authors have provided the dataset and saved it in Huggingface. The data format is in a JSON file with images for the geometries for each exercise, and the evaluation code is on GitHub with a readme that describes the details of the benchmarks. It's easy to run the code

**Ethical Comments:**

The authors mentioned that the geometry problems were sourced from public real-world school tests. I would like to clarify: are these problems taken from actual standardized exams such as the SAT, Gaokao, A-Level, or other official assessments?
If so, are there any copyright considerations or permissions involved in using these problems?

If instead they are general practice problems or derived from public domain resources, a clarification would be helpful for understanding the data provenance and any ethical implications.

**Ethical Considerations:**

Yes, there are ethics concerns that require attention by the authors

**Final Justification:**

Based on the strengths and limitations of the work, I recommend a borderline accept for this paper. It addresses an important and timely need in the evaluation of Multimodal Large Language Models (MLLMs).

The work represents a promising first step. However, to enhance its impact and robustness, I suggest incorporating a broader range of real-world geometry problems, and creating new, original problems that are not publicly available. This would help ensure that the benchmark remains challenging and less likely to overlap with data that MLLMs may have already seen during pretraining.

**Limitations Weaknesses:**

1. Dataset Composition

A significant portion (over 50%) of the dataset is aggregated from existing benchmarks. Although the authors have conducted deduplication, this reliance on pre-existing material somewhat limits the novelty of the benchmark. Furthermore, the scale of the benchmark remains relatively small by today’s LLM standards. To more effectively evaluate the mathematical reasoning capabilities of MLLMs, the dataset would benefit from a greater number of diverse and challenging exercises.


2. Human Baseline Details

The inclusion of a human baseline is a valuable reference point. The paper mentions that the baseline was established using 60 high school students, which is a reasonable starting point. However, the methodology could be strengthened by providing additional details regarding the student cohort (e.g., academic background, region, or grade level), the testing environment, and the protocol for distributing problems among participants. These details would help better assess the reliability of the baseline and the relative difficulty of the problems.

**Strengths Contributions:**

1. Address a big gap: the paper argues for the need for a benchmark focused on solid geometry. This area requires a higher degree of spatial intelligence than plane geometry and is a crucial frontier for evaluating and advancing MLLMs.
2. The benchmarks are in a high-quality design: SOLIDGEO is a well-constructed and valuable resource. It has a scale of 3000+ problems, sourcing for real-world study contexts, and rich annotation including difficulty levels and 8 fine-grained categories.
3. Clear and Reproducible work: The paper is well-written, clearly motivated, and easy to follow. I think the commitment to releasing the dataset and code is commendable and will be of great benefit to the research community.

---

> ### Author Rebuttal · Authors · 2025-07-31
>
> Dear Reviewer Ab3A:
>
> Thank you for your thoughtful and constructive review. We greatly appreciate your recognition of SOLIDGEO's contributions, as well as your valuable suggestions for improving the benchmark. Based on your feedback, we will incorporate the corresponding revisions into the final version of the manuscript to enhance clarity, transparency, and overall quality. Below, we address each of your comments in detail.
>
> > **Q1: Dataset Composition**
>
> We appreciate your comments regarding the dataset's composition and scale, and welcome the opportunity to highlight the novelty of our benchmark and clarify our design choices.
>
> **On Dataset Composition and Novelty**
>
> You've correctly pointed out that a significant portion of SOLIDGEO is sourced from existing datasets. We chose this hybrid approach because high-quality, real-world solid geometry problems are inherently scarce and challenging to collect from scratch. However, we would like to respectfully highlight that the primary novelty of our work lies not just in creating new problems, but in the **pioneering effort to systematically establish and structure the domain of solid geometry reasoning as a distinct evaluation frontier.**
>
> Our key contributions to novelty are:
>
> 1. **Systematic Curation and Unification:** Before our work, solid geometry problems were scattered and overlooked. SOLIDGEO is the first benchmark to identify, meticulously filter, and unify these problems from disparate sources into a single, coherent resource. This process involved significant effort in deduplication, reformatting, and re-annotating every single problem to ensure consistency and quality.
> 2. **Novel Fine-Grained Taxonomy:** A core original contribution of our work is the introduction of the 8-category fine-grained classification for solid geometry reasoning (e.g., *Planar Unfolding and Configuration*, *Multi-view Projection*). This taxonomy is entirely new and provides the community with a crucial tool for in-depth, structured analysis of MLLM capabilities, which was lacking in existing benchmarks.
> 3. **Substantial New Data:** We also contributed 1,376 entirely new, manually verified problems (44.2% of the dataset), which significantly enriches the diversity and complexity of the available questions in this domain.
>
> **On the Scale of the Benchmark**
>
> Regarding the scale of our benchmark, we appreciate your perspective. For this **inaugural version of SOLIDGEO**, our goal was to create a benchmark that is both substantial enough for robust evaluation and practical for the community to use. At 3,113 problems, SOLIDGEO is already a large-scale benchmark in the specialized domain of multimodal math reasoning, comparable to or larger than prominent peers like MathVision (3,040) [1] and MathVerse (2,162) [2]. We deliberately chose this scale to balance statistical significance with evaluation efficiency, as these complex reasoning tasks are highly resource-intensive. We argue that for the contribution presented in this paper, this scale is appropriate and sufficient. That said, we agree with the reviewer that further scaling—particularly with more diverse and challenging original problems—is essential for the benchmark’s long-term value. We view SOLIDGEO as a living benchmark, and are committed to its principled expansion as more high-quality data becomes available in the future.
>
>
>
> > **Q2: Human Baseline Details**
>
> Thank you for your insightful suggestion. We agree that providing additional methodological details can enhance the transparency and reliability of the human baseline evaluation.
>
> To clarify, we recruited **60 Grade 11 students** who had received formal instruction in solid geometry. All participants had no prior exposure to the SOLIDGEO dataset and were not informed of any model performance results to eliminate bias.
>
> For the evaluation, we randomly divided the 3,113 problems in SOLIDGEO into 60 subsets, each containing approximately 51–52 questions. Each student was randomly assigned two subsets, and asked to complete them independently in two separate 150-minute sessions over two days, simulating a real-world testing environment. All answers were written by hand under supervision. Grading was conducted by two experienced mathematics instructors.
>
> We will include these details in the final version of the paper to better support the assessment of the human baseline's reliability.
>
>
>
> > **Q3: Ethical Considerations — Data Provenance and Copyright**
>
> Thank you for raising this important question regarding data provenance and ethical integrity.
>
> We would like to clarify that our newly collected problems are sourced from **general practice exercises and public-domain educational resources**. To ensure compliance with ethical and legal standards, we established a formal collaboration with **TAL**, a leading education technology company specializing in K–12 learning.
>
> Through this partnership, we followed a strict, multi-stage process:
>
> 1. **Source Selection:** We identified suitable practice problems and public domain resources under the guidance of TAL's educational experts.
> 2. **Copyright Verification:** A crucial part of this collaboration was that TAL's team conducted rigorous copyright verification on all selected materials to ensure they were fully cleared for research use.
> 3. **Data Extraction:** Only after this verification did we utilize the Mathpix API to extract the structured content of the problems.
>
> This process ensured that all content included in SOLIDGEO is ethically sourced and free of intellectual property concerns.
>
> To improve transparency, we will include a detailed description of this collaboration and the vetting pipeline in **Appendix D.1 (Data Collection Details)** in the revised version of the paper.
>
>
>
> > **Additional Feedback about SOLIDGEO**
>
> Thank you for this forward-looking suggestion. This aligns perfectly with our long-term vision for SOLIDGEO.
>
> While the current scale of SOLIDGEO is robust for an initial benchmark, we wholeheartedly agree that the **longevity and future impact** of any benchmark depend on its continued evolution. We appreciate your insightful suggestion to incorporate more **original and non-public problems**. This indeed aligns with an important future direction—reducing the risk of potential **data contamination** during model pretraining and evaluation.
>
> Therefore, when we discuss "increasing the dataset's scale" for future work, we are referring specifically to the **strategic addition of high-value, original problems**, not just a simple quantitative increase. Our commitment is to the thoughtful growth of SOLIDGEO, striking a balance between:
>
> 1. **Enhancing problem diversity** with novel, non-public content.
> 2. **Strengthening evaluation integrity** against data contamination.
> 3. **Maintaining evaluation efficiency** for the community.
>
> In summary, we believe the current version of SOLIDGEO is a complete and robust contribution as it stands. Your feedback is valuable in shaping the future roadmap for SOLIDGEO. We will add a discussion of this strategic direction to the 'Limitations and Future Work' section.
>
> **Reference**
>
> [1]. Ke Wang, Junting Pan, Weikang Shi, Zimu Lu, Houxing Ren, Aojun Zhou, Mingjie Zhan, and Hongsheng Li. Measuring multimodal mathematical reasoning with math-vision dataset. Advances in Neural Information Processing Systems, 37:95095–95169, 2024.
>
> [2]. Renrui Zhang, Dongzhi Jiang, Yichi Zhang, Haokun Lin, Ziyu Guo, Pengshuo Qiu, Aojun Zhou, Pan Lu, Kai-Wei Chang, Yu Qiao, et al. Mathverse: Does your multi-modal llm truly see the diagrams in visual math problems? In European Conference on Computer Vision, pages 169–186. Springer, 2024.

---

> > ### Comment · Reviewer_Ab3A · 2025-08-06
> >
> > Thank you to the authors for their detailed responses, which have addressed all my concerns. I recommend accepting this paper.

---

> > > ### Author Response · Authors · 2025-08-06
> > >
> > > Thank you for your positive feedback and support. We are glad that our rebuttal has addressed your concerns. We are also very grateful for your constructive comments, which will help us significantly improve the final manuscript.

---

### Note · Authors · 2025-08-12

Dear Area Chair,

We would first like to thank you for overseeing the review process and all the reviewers for their highly constructive and insightful feedback.

We were greatly encouraged by the supportive initial assessment from all reviewers and the productive discussion phase that followed. This is best exemplified by Reviewer Ab3A, who had initially given a 'Borderline accept' rating, but stated in their final official comment: "Thank you to the authors for their detailed responses, which have addressed all my concerns. I recommend accepting this paper."

Furthermore, we wish to proactively summarize for you how the most critical concerns—particularly the ethical issues raised by Reviewer 1Nsg—have also been fully resolved:

Human Subjects: Our study was conducted under a formal ethical protocol with school approval, consent from guardians alongside assent from the students themselves, and appropriate compensation. The NeurIPS checklist will be corrected to reflect this.

Data Privacy & Copyright: We secured formal authorization from the Zujuan platform for academic use. The released SolidGeo benchmark contains no PII and is published under a standard CC-BY-NC-4.0 license.

We believe our responses have thoroughly addressed all points raised across all reviews. We are confident that SolidGeo is a robust and valuable contribution to the community and thank you for your time and final consideration.

Sincerely,

The Authors of Submission 469

---

### Decision · Program_Chairs · 2025-09-18

**Decision:**

Accept (poster)

**Comment:**

This paper addresses a critical gap in multimodal reasoning by introducing SOLIDGEO, the first large-scale benchmark for solid geometry. The paper received three Accept and one Borderline accept recommendations. While reviewers initially raised minor concerns regarding the human baseline methodology, difficulty labeling process, and reliance on 2D image projections, the authors provided a comprehensive rebuttal that satisfactorily addressed points.  All reviewers agreed the new benchmark is novel, large, and rigorous evaluation, highlighting its value for the community. Given the paper's significant contribution in defining a new, challenging task and providing a high-quality benchmark to drive future research, the AC recommends acceptance.

===== FINAL UPDATE FROM DB Track PCs ====

The final decision for this paper has been taken by the program chairs after consultation with the SACs. All Senior Area Chairs have ranked papers according to the feedback from the AC during the review process. We decided to leave the original meta-review to reflect the opinion of the AC in light of the initial discussions with reviewers and SAC.